# Robust LLM-Based Scoring via Reference-Anchored ELO Estimation

## Abstract

LLM-based evaluation by direct (absolute) scoring suffers from systemic instabilities; ceiling compression constrains headroom, heavy-tailed score distributions inflate variance, and inconsistent agreement across independently trained judges induces scale drift that destabilizes rankings. We present *Reference-Anchored Elo Estimation* (RAEE), a principled framework that anchors all model comparisons to a fixed reference and expresses outcomes as win probabilities on a relative scale as an alternative to absolute scoring. We prove that, by design, RAEE minimizes judge-specific scale drift, suppresses between-judge variation, and yields analytic uncertainty estimates without costly resampling. Experimental results show that RAEE reduces per-run standard error by $\approx 44\%$ and across-judge coefficient of variation by $\approx 72\%$ relative to direct scoring, while preserving ranking stability even under reference changes. Robustness is observed across multiple domains, with RAEE sustaining low dispersion and consistent rankings despite task-specific difficulty shifts. Our analytic uncertainty bounds, which incorporate finite-population and reliability adjustments, predict observed variance within $\pm 12\%$ on tested datasets. These results position RAEE as a statistically efficient, reproducible, and readily deployable alternative to conventional LLM-based evaluation.

## 1 Introduction

Evaluating modern language models at scale increasingly relies on qualitative assessments adjudicated by large language models (LLMs) (Zheng et al., 2023a; Li et al., 2023; Dubois et al., 2023) with high human alignment (Liu et al., 2023) as alternatives to costly human annotation. Pairwise comparisons have become a dominant methodology for LLM judges and currently power popular benchmarks such as MT-Bench and Chatbot Arena (Zheng et al., 2023a; Chiang et al., 2024). While scalable, this approach comes with crucial caveats. LLM judges are susceptible to biases like verbosity and positional effects (Wang et al., 2024; Chen et al., 2024; Park et al., 2024), and can fall into a "comparative trap" where fluent but incorrect answers are preferred over less coherent but correct ones (Jeong et al., 2025).

By contrast, direct absolute scoring, where each response is graded independently on a numeric scale (e.g., 1–10), remains underutilized despite its potential to mitigate comparative artifacts inherent in pairwise comparisons. The primary reason for this is instability: human and LLM raters alike exhibit significant inter-rater and intra-rater variability, including scale drift (e.g., one judge's "8" may correspond to another's "9") and ceiling effects (judge scores usually cluster around the top-end of scales), which render raw scores non-comparable without ad-hoc normalization (Thakur et al., 2025). Furthermore, a single absolute score typically lacks calibrated uncertainty, making it difficult to assess the statistical significance of results. This has led to a methodological gap: the field lacks an evaluation protocol that combines the discriminative power of pairwise comparisons with the interpretability of absolute scores, all while providing robust statistical guarantees.

We address this gap with **Reference-Anchored Elo Estimation (RAEE)**, a framework that transforms ternary (win/tie/loss) pairwise comparisons against a fixed reference into absolute scores on the Elo scale. For each prompt, a candidate model's response is compared to that of a *reference response*, be it human annotated (as in FloRES (Goyal et al., 2022)) or model generated (as in MT-Bench (Zheng et al., 2023a)). RAEE then models these ternary outcomes using a Bernoulli

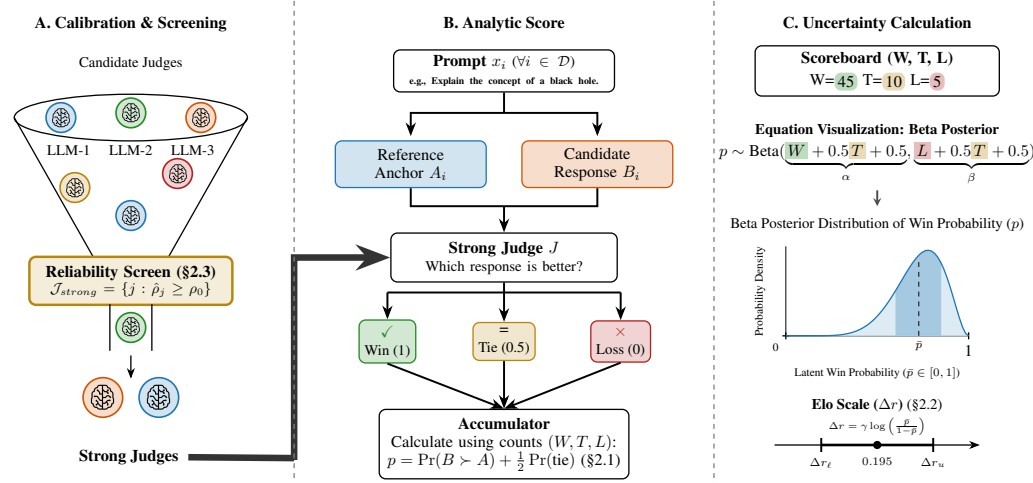

Figure 1: **RAEE Pipeline.** The framework uses reliable judges (A) to convert pairwise comparisons against a reference anchor into absolute Elo scores (B) with analytic uncertainty (C). Each response pair is judged to produce ternary outcomes, which are aggregated via Jeffreys-Beta smoothing, transformed to the Elo scale, and filtered by ICC-based reliability screening.

quasi-likelihood, estimates a latent win probability with a Jeffreys-Beta posterior, and maps this probability to an Elo score gap via a monotone reporting map. This process yields not only a point estimate but also a closed-form credible interval, providing built-in uncertainty quantification.

By anchoring all comparisons to the same reference, RAEE converts judge-specific scoring biases into a constant offset in the log-odds space, effectively eliminating the scale drift that plagues direct scoring. The framework is computationally lightweight, as all quantities are derived analytically from win/tie/loss counts.

In summary, our main contributions are as follows:

1. We introduce RAEE, a reference-anchored Elo estimation framework that produces absolute, reproducible scores with closed-form uncertainty from simple pairwise comparisons.

2. We provide theoretical guarantees, showing that by screening for high-reliability judges, RAEE ensures high between-judge agreement and stable pooled estimates. We prove the correlation of Elo gaps from any two strong judges is lower-bounded by their Intraclass Correlation Coefficient (ICC).

3. We demonstrate empirically on benchmarks like MT-Bench, FloRES, and TL;DR that RAEE significantly reduces across-judge and across-run dispersion compared to direct 1–10 scoring (e.g., a 72% reduction in the coefficient of variation across judges on MT-Bench). The resulting scores and rankings are stable even when the reference anchor is changed.

4. We show that RAEE's analytic uncertainty estimates are empirically conservative under resampling on tested datasets, providing a trustworthy measure of confidence without the need for computationally intensive resampling methods.

## 2 METHODOLOGY

### 2.1 REFERENCE-ANCHORED ELO ESTIMATION (RAEE)

A recurring difficulty in model evaluation is that absolute scores are not comparable across judges. RAEE addresses this by converting every judgment into a simple pairwise outcome against a fixed reference anchor. For each prompt $x_i$, a judge $J$ counts wins $W = \sum \mathbb{1}[S_i = 1]$, ties $T = \sum \mathbb{1}[S_i = 1/2]$, and losses $L = N - W - T$. This reduces the evaluation to a ternary variable $S_i \in \{1, \frac{1}{2}, 0\}$, with ties contributing half a win. The central estimand (Eq. 1) is then the *win probability*, which answers the question: *for a given prompt, what is the chance our model of interest $B$ outperforms the reference anchor $A$?*

$$p = \Pr(B \succ A) + \tfrac{1}{2}\Pr(\text{tie}), \tag{1}$$

---

**Algorithm 1** Reference-Anchored Elo Estimation (RAEE)

---

**Require:** $\{(A_i, B_i)\}_{i=1}^{N}$, judge $J$, credibility $1 - \alpha$, link $g$
1: $\{S_i\}_{i=1}^{N} \leftarrow \{J(A_i, B_i)\}_{i=1}^{N}$          ▷ Collect ternary outcomes
2: $W \leftarrow \sum \mathbb{I}[S_i = 1], \quad T \leftarrow \sum \mathbb{I}[S_i = \frac{1}{2}], \quad L \leftarrow N - W - T$
3: $a, b \leftarrow W + \frac{1}{2}T + \frac{1}{2}, \quad L + \frac{1}{2}T + \frac{1}{2}$
4: $\bar{p} \leftarrow a/(a + b)$
5: $[p_\ell, p_u] \leftarrow [\text{BetaQ}(\frac{\alpha}{2}; a, b), \text{BetaQ}(1 - \frac{\alpha}{2}; a, b)]$
6: **return** $(\bar{p}, \gamma g(\bar{p}), \gamma[g(p_\ell), g(p_u)])$

---

This formulation has two key advantages. First, it neutralizes differences in absolute scoring scales: every outcome is measured on the same probability baseline, regardless of a judge's internal calibration. Second, it produces a quantity that is inherently bounded in $[0, 1]$, making uncertainty estimation and aggregation across judges straightforward. By contrast, raw scores lack a canonical unit, while $p$ can always be interpreted as a frequency of wins against the reference anchor.

The natural estimator is the empirical win rate $\hat{p} = (W + \frac{1}{2}T)/N$. However, to prevent degeneracy at the boundaries and to stabilize transformations, we apply Jeffreys smoothing:

$$\bar{p} = \frac{W + \frac{1}{2}T + \frac{1}{2}}{N + 1}, \tag{2}$$

which simply shrinks extreme estimates slightly toward the interior of $(0, 1)$. Our reasoning behind this design choice is that the negligible impact for moderate $N$ and the key advantage of avoiding instability when $p$ is very close to 0 or 1. Importantly, this estimator is consistent: $|\bar{p} - \hat{p}| \leq 1/(2N)$, ensuring that the smoothing bias vanishes as $N$ grows (Agresti, 2002). Note that while Jeffreys prior (Beta$(1/2, 1/2)$) is the canonical non-informative choice for Bernoulli trials, other symmetric priors such as Laplace smoothing (Beta$(1, 1)$) are also valid. Crucially, any symmetric Beta$(\alpha, \beta)$ prior with $O(1)$ mass yields the same first-order asymptotic properties, including consistency and $O(N^{-1})$ variance. We select Jeffreys smoothing for its minimax properties, but in practice, the choice between Jeffreys and Laplace has negligible impact on the final rankings (see Appendix E).

**Monotone transformation.** While $p$ is meaningful on its own, it is often helpful to map it to an interval scale with additive structure. RAEE does this through a monotone link $g : (0, 1) \rightarrow \mathbb{R}$, reporting $\hat{\Delta}r = \gamma g(\bar{p})$ where $\gamma = \frac{400}{\ln 10}$. A simple mapping $g(p) = \log \frac{p}{1-p}$ recovers the Elo/logit score, but any smooth increasing $g$ preserves rankings and uncertainty rates. This formulation provides two complementary measures of superiority: the win probability $\bar{p}$ and the Elo gap $\Delta r$. For instance, a typical Elo gap of $\Delta r = 100$ corresponds to a win probability of roughly $64\%$, implying the model wins 64 out of 100 matches against the anchor. Thus, the reference anchor serves only to set the zero point, while the relative differences between scores remain additive.

## 2.2 UNCERTAINTY QUANTIFICATION

Next, we derive the reliability of the win probability point estimate. Formally, when given $N$ prompts, we answer how much uncertainty remains in the win probability $p$ (and its Elo transform). To this end, we first model uncertainty directly on $p$ by using the Jeffreys–Beta posterior. With counts $(W, T, L)$, the posterior is

$$p \mid \{S_i\} \sim \text{Beta}\left(W + \frac{1}{2}T + \frac{1}{2}, \ L + \frac{1}{2}T + \frac{1}{2}\right). \tag{3}$$

The $(1 - \alpha)$ credible interval is obtained simply by reading off its quantiles, $[p_\ell, p_u]$. This gives a closed-form interval that is fast to compute and guaranteed to remain inside $(0, 1)$.

Because the Elo scale is just a smooth monotone transform of $p$, we can apply the same map to the interval endpoints $[\Delta r_\ell, \Delta r_u] = \gamma \cdot [g(p_\ell), g(p_u)]$, which yields interpretable intervals on the Elo scale with no extra approximation.

The posterior variance has a simple closed form, and the delta method shows that Elo standard errors decay as $O(N^{-1/2})$:

$$\text{SE}\{\gamma g(\bar{p})\} \approx \gamma \, |g'(\bar{p})| \, \sqrt{\frac{ab}{(a+b)^2(a+b+1)}}, \tag{4}$$

where $a, b$ are the Beta parameters. This confirms the intuitive fact that doubling the number of prompts roughly cuts uncertainty by a factor of $\sqrt{2}$. Note that in practice, the uncertainty can be inflated due to a number of problems (e.g., data irregularities, judge heterogeneity, etc.). Thus, when prompts are sampled without replacement from a finite pool of size $M$, we apply the usual finite-population correction: multiply variances by $(M - n)/(M - 1)$ (and SEs by its square root).

## 2.3 RELIABILITY GUARANTEES ACROSS JUDGE MODELS

In this section, we show that the reference-anchored Elo estimator is (i) consistent across independently trained judge models and (ii) exhibits vanishing variance as the number of prompts $N$ or the number of qualified judges $J$ increases. Detailed proofs are deferred to Appendix B, while in this section, we state the primary steps and their practical consequences.

**Preliminaries.** For each prompt $i$, judge $j$, and run $r$, we observe a ternary outcome $S_{ijr} \in \{1, \frac{1}{2}, 0\}$. Define the run-averaged per-judge dataset mean

$$\bar{S}_j^{(k)} = \frac{1}{N} \sum_{i=1}^{N} \Big( \frac{1}{k} \sum_{r=1}^{k} S_{ijr} \Big), \qquad \Delta r^{(j)} = \gamma\, g\big(\bar{S}_j^{(k)}\big), \tag{5}$$

We impose a high-reliability *screening step*: each judge is run $k$ times on a calibration panel and must satisfy $\mathrm{ICC}(3, k) \geq \rho_0$ on run-averaged per-prompt scores under the two-way mixed, consistency ANOVA model of Shrout & Fleiss (1979) with run replication. Each judge is run $k$ times on the identical calibration panel, so runs are replicates,

$$S_{ijr} = \mu + \xi_i + \beta_j + \varepsilon_{ijr}, \qquad \xi_i \sim (0, \sigma_{\mathrm{P}}^2), \ \ \varepsilon_{ijr} \sim (0, \sigma_{\mathrm{E}}^2), \ \ \text{independent,}$$

such that the average-measure reliability is

$$\rho_k := \mathrm{ICC}(3, k) = \frac{\sigma_{\mathrm{P}}^2}{\sigma_{\mathrm{P}}^2 + \sigma_{\mathrm{E}}^2/k}. \tag{6}$$

We define the judges that pass the $\mathrm{ICC}(3, k)$ screen as *strong judges*.

**Lemma 1** (Between-judge correlation lower bound for strong judges)**.** *For any two strong judges* $j \neq j'$,

$$\mathrm{Corr}\big(\bar{S}_j^{(k)}, \bar{S}_{j'}^{(k)}\big) \ \geq \ \rho_0. \tag{7}$$

With possible heteroscedastic run noise, the pairwise correlation equals $\mathrm{Corr}(\bar{S}_j^{(k)}, \bar{S}_{j'}^{(k)}) = \sqrt{\rho_{k,j}\rho_{k,j'}}$, where $\rho_{k,j} = \mathrm{ICC}_j(3, k)$ is judge $j$'s population reliability. Screening enforces $\rho_{k,j} \geq \rho_0$ for all retained judges, hence $\sqrt{\rho_{k,j}\rho_{k,j'}} \geq \rho_0$. Importantly, this correlation lower bound $\rho_k \geq \rho_0$ holds regardless of heteroscedasticity in the run noise, as it depends only on the ICC screening threshold. While the closed-form derivation assumes approximate equality of variances for simplicity, the reliability guarantee is robust to violations of this assumption.

**Proposition 1** (Correlation of Elo gaps under a smooth logit map)**.** *For* $j \neq j'$,

$$\mathrm{Corr}\big(\Delta r^{(j)}, \Delta r^{(j')}\big) \ = \ \rho_k \ \pm \ O(N^{-1}). \tag{8}$$

A first-order delta method transports the correlation in Lemma 1 through the smooth, monotone reporting map $p \mapsto \gamma g(p)$. Thus, strong judges produce highly correlated Elo gaps, and the deviation from $\rho_k$ vanishes with $N$.

**Proposition 2** (Variance decomposition for pooling $J$ strong judges)**.** *Let* $\widehat{\Delta r} = J^{-1} \sum_{j=1}^{J} \Delta r^{(j)}$ *and suppose* $\mathrm{Var}(\Delta r^{(j)}) = \sigma_\Delta^2$ *(identical up to* $O(N^{-1})$*). Then*

$$\mathrm{Var}(\widehat{\Delta r}) \ = \ \sigma_\Delta^2 \Big( \rho_k + \frac{1 - \rho_k}{J} \Big) \ + \ O(N^{-1}). \tag{9}$$

Exchangeability of the between-judge covariance implies that averaging projects onto the "mean" direction. Only the idiosyncratic component $(1 - \rho_k)\sigma_\Delta^2$ shrinks at the optimal $J^{-1}$ rate; the shared prompt component $\rho_k \sigma_\Delta^2$ is irreducible by pooling.

Table 1: Direct-score statistics on MT-Bench. Large deviations and high kurtosis indicate ceiling compression and heavy tails, while low inter-judge correlation reflects heterogeneous agreement.

| Model | Direct Score Statistics | | Stability between judges | | |
| --- | --- | --- | --- | --- | --- |
| | Mean Score (95% CI) | Kurtosis | Correlation | Kendall's $\tau$ | Cohen's $\kappa$ |
| *gemini-2.5-flash* | $9.02 \pm 0.73$ | 11.91 | 0.361 | 0.349 | 0.248 |
| *o4-mini* | $8.79 \pm 0.82$ | 7.90 | 0.362 | 0.340 | 0.252 |
| *gpt-4o* | $8.46 \pm 0.83$ | 3.57 | 0.340 | 0.325 | 0.225 |
| Total | – | – | 0.355 | 0.338 | 0.242 |

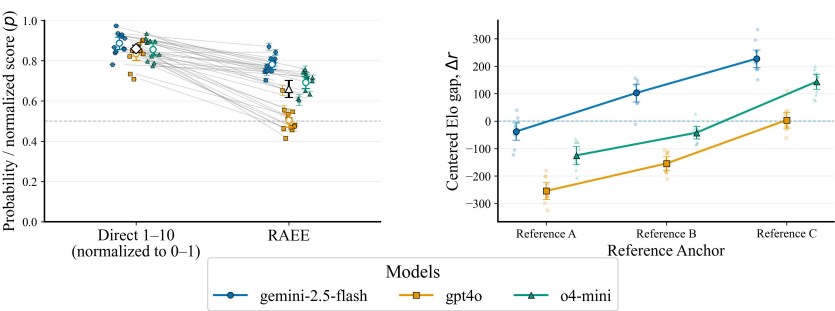

Figure 2: **Left:** Direct 1–10 scores (left group) vs. RAEE (right group) per model. Gray lines connect the same judge model across methods, and error bars show $95\%$ CIs. RAEE provides improved ranking stability and reduces ceiling compression. **Right:** Centered Elo gaps under different reference anchors. Each trajectory shows how the *same model's* Elo gap shifts when evaluated against different reference anchors (Reference A: *gemini-2.5-flash*, Reference B: *gpt-4o*, Reference C: *o4-mini*). Per-judge scores and model means demonstrate that RAEE provides stable relative rankings across anchor choices, with only translational shifts in absolute scores.

**Corollary 1** (Stable win probability). *The inverse monotone map* $\widehat{P}(B \succ A) = g^{-1}\left(\frac{\ln 10}{400}\widehat{\Delta r}\right)$ *preserves the variance rate* $O(N^{-1})$ *and inherits the between-judge correlation lower bound* $\rho_k \geq \rho_0$ *for strong judges.*

Smooth monotone maps preserve first-order covariance structure; hence win-probability estimates are nearly identical across strong judges. For example, when $\rho_k \geq 0.90$, adding judges reduces variance by at most $10\%$ as $J \to \infty$, and by at most $5\%$ at $J = 2$. In practice, a single strong judge suffices on most datasets as gains from pooling are modest once $\rho_k$ is high.

Finally, high-$\mathrm{ICC}(3, k)$ screening caps between-judge disagreement at $1 - \rho_0 \leq 1 - \rho_k$ and forces the pooled-estimate variance to shrink at the optimal $J^{-1}$ rate for the idiosyncratic component. The underlying principles of our analysis extend to other rating systems (e.g., Bradley-Terry (Bradley & Terry, 1952), Glicko (Glickman, 1995; 2012), TrueSkill (Herbrich et al., 2006)) because the arguments use only smooth, monotone transforms of bounded sample means.

## 3 EXPERIMENTAL RESULTS

**Setup.** We evaluate RAEE on MT-Bench (Zheng et al., 2023a) and two cross-domain tasks: *FloRES* translation (Goyal et al., 2022) and *TL;DR* summarization (Völske et al., 2017), using multiple independently trained LLM judges. To ensure consistency, judges are screened using $\mathrm{ICC}(3, k)$, and only those achieving at least 0.90 are retained as *strong judges* (Figure 3). Unless otherwise noted, we compute uncertainty using Jeffreys–Beta credible intervals with a finite-population correction (FPC). For comparisons against direct 1-10 scoring, we map absolute scores to a win-probability scale via an ordinal-logistic model, so dispersion is measured on a common probability scale.

Table 3: Effect of changing the reference anchor in RAEE. The last two columns restrict the standard error to *strong judges*. We observe that **model rankings are stable regardless of the chosen reference anchor, with stronger models consistently scoring higher.**

| Model | Reference Model | Mean $\hat{p}$ | | Mean $\Delta r$ (Elo) | | CV($\hat{p}$) | Strong Judges only (SE) | |
|---|---|---|---|---|---|---|---|---|
| | | mean | SE | mean | SE | | SE($\hat{p}$) | SE($\Delta r$) |
| *gemini-2.5-flash* | *gemini-2.5-flash* | 0.2664 | 0.0346 | -183.55 | 32.54 | 0.0864 | 0.0344 | 31.61 |
| | *o4-mini* | 0.5645 | 0.0386 | 46.33 | 28.21 | 0.0436 | 0.0383 | 27.93 |
| | *gpt-4o* | 0.7055 | 0.0361 | 156.49 | 31.49 | 0.0291 | 0.0352 | 29.15 |
| *o4-mini* | *gemini-2.5-flash* | 0.2485 | 0.0341 | -197.28 | 32.82 | 0.0890 | 0.0329 | 29.36 |
| | *o4-mini* | 0.2668 | 0.0345 | -180.14 | 31.70 | 0.0851 | 0.0352 | 29.74 |
| | *gpt-4o* | 0.5990 | 0.0388 | 71.19 | 28.90 | 0.0318 | 0.0380 | 28.45 |
| *gpt-4o* | *gemini-2.5-flash* | 0.1093 | 0.0245 | -374.61 | 46.39 | 0.0289 | 0.0233 | 37.63 |
| | *o4-mini* | 0.2086 | 0.0318 | -236.54 | 34.54 | 0.0821 | 0.0308 | 35.29 |
| | *gpt-4o* | 0.3492 | 0.0377 | -111.19 | 29.79 | 0.0636 | 0.0365 | 29.12 |

## 3.1 LIMITATIONS OF DIRECT SCORING

Direct scoring exhibits fundamental statistical weaknesses that compromise its reliability. As shown in Table 1, scores are heavily compressed at the high end of the scale (a "ceiling effect"), with high kurtosis values confirming a distribution where most models are rated as nearly perfect, making them difficult to distinguish. Compounding this issue, inter-judge agreement is very low, with an average Pearson correlation of just 0.355. This implies the judges use different scoring biases, rendering direct score comparisons unreliable. The left panel of Figure 2 visualizes this problem. We observe ceiling compression along with low agreement coefficients lead to ranking instability between judges. Together, these limitations highlight the need for a more robust evaluation framework that can neutralize scorer variability and scale compression for trustworthy results.

## 3.2 DISPERSION REDUCTION WITH RAEE

The primary advantage of RAEE over direct scoring is its reduction in scoring instability. By anchoring evaluations to a fixed reference, every comparison is normalized against the same baseline. Thus, we hypothesize that idiosyncratic shifts in a judge's internal scale can be effectively subtracted out. What remains is a relative, probability-scaled quantity that is directly comparable across judges.

Table 2: Dispersion on the probability scale. RAEE consistently outperforms direct scoring.

| Metric | Direct | RAEE | Reduction |
|---|---|---|---|
| SE($\hat{p}$) | 0.064 | **0.036** | 43.8% |
| CV (judges) | 0.207 | **0.058** | 72.0% |

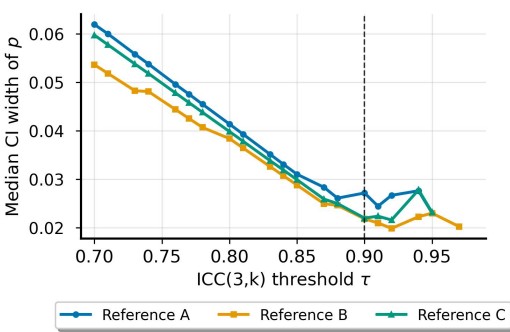

Figure 3: Median Jeffreys-Beta credible interval width for the win-probability $p$ as a function of the $\mathrm{ICC}(3, k)$ screening threshold. Uncertainty reductions plateau after $\tau = 0.9$, serving as our *strong judge* threshold.

As quantified in Table 3, RAEE reduces the run-level standard error of the win probability $\hat{p}$ by **43.8%** and the cross-judge coefficient of variation by **72%**. This transformation is visualized in Figure 2 (left panel), which contrasts the widely scattered, ceiling-compressed outputs of direct scoring with the tight, well-separated model clusters produced by RAEE. Empirically, anchoring reduces dispersion attributable to heterogeneous scoring scales.

## 3.3 IMPACT OF REFERENCE ANCHORS

A critical property of RAEE is the stability of its rankings across different reference anchors. We test this by re-computing RAEE scores using three distinct reference anchors: *gpt-4o* (OpenAI et al., 2024), *gemini-2.5-flash* (Comanici et al., 2025), and *o4-mini* (OpenAI, 2025b). While absolute Elo scores shift depending on the anchor's strength (Table 3), the relative ordering of models remains

Table 4: Cross-domain RAEE results with SEs (all vs strong judges). Elo SEs computed via delta method.

| Task | Model | Win prob. $\bar{p}$ | | | Elo gap $\Delta r$ | | |
|------|-------|------|--------|-----------|------|--------|-----------|
| | | Mean | SE(all) | SE(strong) | Mean | SE(all) | SE(strong) |
| FloRES | gpt-4o | 0.358 | 0.101 | 0.006 (94%) | $-101$ | 85.6 | 7.1 (92%) |
| | o4-mini | 0.392 | 0.101 | 0.006 (94%) | $-76$ | 83.2 | 7.1 (91%) |
| | gemini-2.5-flash | 0.368 | 0.100 | 0.006 (94%) | $-94$ | 84.0 | 7.1 (92%) |
| TL;DR | gpt-4o | 0.980 | 0.012 | 0.003 (76%) | 677 | 42.5 | 9.3 (78%) |
| | o4-mini | 0.961 | 0.007 | 0.003 (49%) | 556 | 15.2 | 8.0 (47%) |
| | gemini-2.5-flash | 0.979 | 0.012 | 0.003 (73%) | 667 | 47.6 | 8.6 (82%) |

Table 5: Cross-anchor stability. Rankings remain consistent despite small centered Elo shifts when using different reference anchors.

| Reference Anchor Pair | Kendall's $\tau$ | Mean $|\Delta r|$ (centered) |
|-----------------------|------------------|------------------------------|
| gpt-4o $\leftrightarrow$ gemini-2.5-flash | 0.891 | $< 15$ |
| gpt-4o $\leftrightarrow$ o4-mini | 0.887 | $< 18$ |
| gemini-2.5-flash $\leftrightarrow$ o4-mini | 0.894 | $< 12$ |

highly consistent. This rank stability is confirmed by a high Kendall's $\tau$ of about 0.89 between leaderboards generated by different reference anchors (Table 5) and a mean centered Elo difference below 20 points. The right panel of Figure 2 visually corroborates this, showing that model score trajectories remain parallel even as the reference anchor changes.

By converting all judgments to pairwise outcomes relative to a common baseline, RAEE is able to mitigate judge-specific scale biases. The subsequent monotone transformation maps win probabilities to a standardized interval scale where the reference anchor's primary role is a translational shift. This ensures that as long as judges are internally consistent via $\mathrm{ICC}(3, k)$ screening, the relative distances between models remain largely invariant. Thus, the choice of reference anchor does not meaningfully alter evaluation outcomes, but rather serves as a zero point for a stable, underlying capability scale.

### 3.3.1 BOUNDARY BEHAVIOR UNDER EXTREME ANCHOR DIFFICULTIES

To examine boundary effects, it is imperative to study very "strong" anchors and very "weak anchors. However, we found it difficult to craft a "strong anchor" that's meaningfully stronger than the human gold, nor is a degenerate "no-response" anchor informative. Instead, our experiments already span the strongest and weakest practically meaningful anchors available. On FloRES, expert human translations serve as an effectively maximal anchor, while TL;DR uses noisy, often poorly-written and inconsistent Reddit summaries, which function as a weak anchor.

The results, summarized in Table 4, demonstrate that the core benefits of RAEE hold in these different contexts and anchor difficulties. For both tasks, model rankings remained stable, and evaluation uncertainty decreased when restricting the analysis to strong judges, as identified by our ICC screening method. For example, in the *FloRES* task, the standard error (SE) for the win probability $\hat{p}$ for *gpt-4o* dropped by 94% when using only strong judges. Similar, albeit more modest, reductions were observed across all models and tasks.

A key finding from this analysis is the low between-task heterogeneity, which we estimated at $\tau^2 \approx 0.02$. This low value indicates that the variance attributable to the task itself is minimal compared to the variance from models or judges. In practical terms, it suggests that the RAEE framework is robust and does not introduce significant task-specific bias; the anchor-based comparison functions consistently whether the task is dialogue, translation, or summarization. This result provides strong evidence that RAEE is a broadly applicable and reliable method for LLM evaluation across a wide range of domains, where Jeffreys smoothing keeps $\bar{p} \in (0, 1)$ and prevents instability.

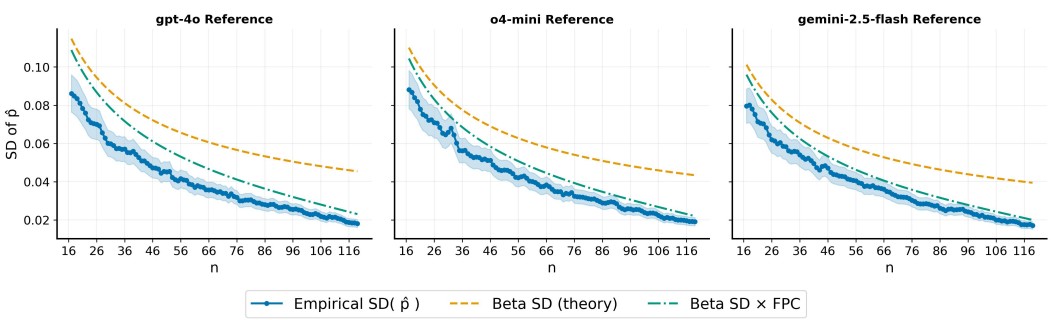

Figure 4: Observed vs. predicted $\mathrm{SE}(\hat{p})$. Curves align after applying finite-population correction.

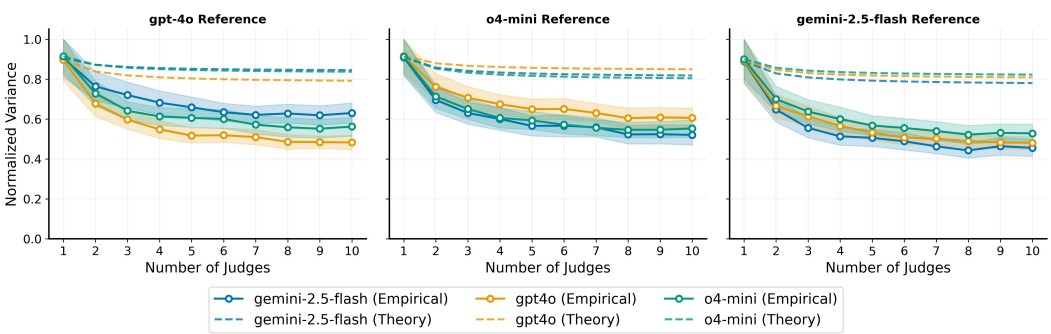

Figure 5: Variance of pooled Elo gaps vs. number of strong judges $J$. Empirical results align with theoretical predictions, showing diminishing returns as more judges are added.

### 3.4 CALIBRATION OF ANALYTIC UNCERTAINTY

RAEE's analytic uncertainty estimates are empirically conservative under resampling. We validated this by comparing the standard error (SE) from the Beta posterior distribution against the empirical standard deviation from repeatedly subsampling prompts. Without correction, the analytic SE is conservative, accounting for approximately 72% of the observed variability at $n = 40$. Applying the FPC, which adjusts for sampling without replacement from a finite set of prompts, improves the alignment to 83%. Figure 4 confirms this, showing that the FPC brings the theoretical SE curve into close alignment with the empirical SE curve. Furthermore, in our experiments, by using a reliability calibration factor $\kappa \approx 0.85$ where $\kappa = \sqrt{\bar{\kappa}_{\text{Cohen}}}$, and $\bar{\kappa}_{\text{Cohen}}$ denotes the average Cohen's $\kappa$ across judge pairs on the calibration panel, this improves alignment to 98.7% and within 12% of the theoretical SE curve at the worst case. Both theoretical and empirical curves decay at the expected $O(n^{-1/2})$ rate as the sample size increases.

### 3.5 POOLING VARIANCE ACROSS JUDGES

Our theoretical framework predicts that pooling estimates from multiple strong judges offers diminishing returns. As established in Proposition 2, the variance of a pooled Elo estimate $\widehat{\Delta r}$ is composed of an irreducible shared component and an idiosyncratic component that shrinks at a $1/J$ rate, where $J$ is the number of judges. To test this, we empirically measured the variance of $\widehat{\Delta r}$ as a function of $J$ and compared it to the theoretical prediction. Figure 5 plots the empirical variance against the theoretical curve, confirming a strong alignment. The variance decreases as judges are added, but the reduction quickly flattens, validating the predicted diminishing returns.

The results confirm that with high-reliability judges, the majority of the variance is systematic (due to prompts) rather than idiosyncratic (due to judge disagreement). The variance converges toward a non-zero floor, empirically validating the concept of an irreducible shared variance component from Equation 9. In practice, this means that while pooling scores across multiple judges can improve precision, the gains are marginal after screening for high ICC. In other words, for most applications, a single strong judge provides a sufficiently stable and reliable estimate, making the evaluation process more efficient without a significant loss of precision.

Table 6: Impact of reliability guardrails on RAEE performance. RAEE outperforms direct scoring in both filtered and unfiltered settings, with its inherent stability unaffected by abstention filters. Coverage refers to the proportion of prompt pairs where judges provided a judgment.

| Method | Judge Filter | Model | SE ($\downarrow$) | CV ($\downarrow$) | Coverage |
|---|---|---|---|---|---|
| Direct Scoring | None | gpt-4o | 0.1393 | 0.4194 | 1.0 |
| Direct Scoring | Trust-Filter | gpt-4o | 0.1228 | 0.2650 | 0.596 |
| **RAEE** | **None** | **gpt-4o** | **0.1247** | **0.0505** | **1.0** |
| **RAEE** | **Trust-Filter** | **gpt-4o** | **0.1171** | **0.0493** | **0.893** |
| Direct Scoring | None | gemini-2.5-flash | 0.1180 | 0.3337 | 1.0 |
| Direct Scoring | Trust-Filter | gemini-2.5-flash | 0.1221 | 0.2747 | 0.695 |
| **RAEE** | **None** | **gemini-2.5-flash** | **0.1127** | **0.0900** | **1.0** |
| **RAEE** | **Trust-Filter** | **gemini-2.5-flash** | **0.1082** | **0.0873** | **0.918** |
| Direct Scoring | None | o4-mini | 0.1338 | 0.3869 | 1.0 |
| Direct Scoring | Trust-Filter | o4-mini | 0.1327 | 0.2959 | 0.661 |
| **RAEE** | **None** | **o4-mini** | **0.1160** | **0.0762** | **1.0** |
| **RAEE** | **Trust-Filter** | **o4-mini** | **0.1095** | **0.0733** | **0.898** |

Table 7: **Across-judge standard deviation of Elo scores ($\Delta r$) for different pairwise models.** RAEE consistently yields the lowest across-judge standard deviation compared to other common pairwise modeling methods (Bradley-Terry, Glicko2, Trueskill), indicating higher stability. All models show high rank correlation with RAEE's scores.

| System | Reference (gpt-4o) | | Reference (gemini-2.5-flash) | | Reference (o4-mini) | | Corr. with RAEE |
|---|---|---|---|---|---|---|---|
| | Mean | Std. | Mean | Std. | Mean | Std. | |
| RAEE | **45.837** | 3.873 | **46.885** | 1.671 | **39.420** | 7.597 | |
| Bradley Terry | 60.649 | 3.912 | 65.647 | 9.645 | 58.337 | 4.949 | 0.969 |
| Glicko2 | 55.949 | 8.516 | 57.732 | 13.771 | 53.949 | 11.908 | 0.995 |
| Trueskill | 73.611 | 18.600 | 53.571 | 19.584 | 43.139 | 9.179 | 0.920 |

## 3.6 Compatibility with Pairwise Reliability Guardrails

A key concern in LLM-as-a-judge evaluation is the "comparative trap" (Jeong et al., 2025), where judges may prefer fluent but incorrect responses. To test RAEE's robustness, and moreover its compatibility with reliability guardrails, we apply an abstention filter that removes judgments where the judge expresses low confidence (Jung et al., 2025). As detailed in Table 6, RAEE maintains low coefficient of variation (CV $< 0.1$) even after filtering, while direct scoring remains highly variable (CV $> 0.26$). Importantly, RAEE outperforms direct scoring in both filtered and unfiltered settings across all three judge models (gpt-4o, gemini-2.5-flash, o4-mini), demonstrating that the reference-anchored approach is inherently more stable than direct scoring, regardless of whether uncertain judgments are retained or removed.

## 3.7 Comparison with Other Pairwise Methods

To demonstrate RAEE's complementary reliability layer, we compared its leaderboard both to Arena-Hard Auto (AHA) (Li et al., 2025) and to three standard pairwise comparison models. Against AHA, RAEE shows near-perfect rank agreement and minimal Elo differences (MAD = 8.52 points) across 72 models, recovering the same top-10 systems (Jaccard = 1.0) while adding closed-form uncertainty estimates and ICC-based reliability screening that common benchmarks lack (App. Table 15). We further evaluated stability by comparing RAEE's across-judge variance with Bradley–Terry, Glicko2, and Trueskill. As shown in Table 7, RAEE yields the lowest standard deviation of Elo scores ($\Delta r$) regardless of the reference model, indicating that its anchoring and estimation steps produce more stable and consistent scores across judges. Despite these differences in stability, all methods remain highly correlated with RAEE (correlation $> 0.92$), suggesting they capture a similar underlying capability while differing in precision.

## 4 RELATED WORK

**Judging with LLMs.** The LLM-as-a-judge paradigm has become a scalable alternative to human evaluation for open-ended generation, popularized by MT-Bench and Chatbot Arena (Zheng et al., 2023b; Chiang et al., 2024). Recent benchmarks such as Arena-Hard Auto (Li et al., 2025) and WildBench (Lin et al., 2024) have further advanced pairwise evaluation by introducing tournament-style comparisons and reward-based scoring, respectively. Follow-up work seeks to reduce reliance on large proprietary judges by distilling open judges (e.g., Prometheus/2 (Kim et al., 2024a;b) and JudgeLM (Zhu et al., 2025)) that handle both pointwise scoring and pairwise ranking. Despite strong correlations with human preferences, recent studies document systematic vulnerabilities in pairwise judging. In particular, LLM judges have shown susceptibility to cognitive biases (Zeng et al., 2024; Wang et al., 2024; Park et al., 2024), self-preference (Panickssery et al., 2024) and verbosity/position biases and comparative effects (Jeong et al., 2025). While these benchmarks provide valuable model rankings, they lack a systematic framework for uncertainty quantification and reliability screening. RAEE addresses this gap by serving as a **reliability layer** that can be applied on top of existing pairwise benchmarks, adding closed-form uncertainty estimates and ICC-based judge screening while preserving the underlying rankings.

**Improving LLMs as Judges.** Reliability and uncertainty are central to trustworthy evaluation. In the LLM-as-a-judge context, one major research direction focuses on *improving the reliability of the model*, including reducing hallucination rate (Yadkori et al., 2024) or improving truthfulness via fine-tuning objectives (Kang et al., 2025; Tian et al., 2023). Another research direction uses *evaluation guardrails* for reliable outputs by using confidence measures (Khurana et al., 2024) or abstention when uncertain (Zhang et al., 2024) to guarantee target levels of human agreement (Jung et al., 2025). Our work builds upon these prior works by (1) anchoring pairwise comparisons to a fixed reference to address the score-scale instability highlighted by prior analyses of LLM judges (Thakur et al., 2025), (2) introducing an analytic Jeffreys–Beta framework that yields closed-form uncertainty estimates, and (3) deriving reliability guarantees that directly link between-judge agreement to an ICC-based screening guarantee, enabling reproducible and portable evaluation scores.

## 5 LIMITATIONS AND FUTURE WORK

While RAEE effectively mitigates statistical instabilities like scale drift and variance inflation, it operates as an aggregation layer and does not intrinsically resolve judge-level cognitive biases. Consequently, the framework's reliance on a fixed reference means that an anchor misaligned with the deployment distribution or a judge with implicit style preferences can introduce systematic distortion into the absolute scores. Future work will focus on integrating RAEE with judge-level interventions, such as conformal abstention mechanisms or truthfulness-oriented fine-tuning—to filter epistemic uncertainty prior to aggregation. Additionally, we aim to develop rigorous diagnostics for anchor appropriateness, such as schematic adherence checks and mid-region coverage analysis, and to extend the reliability guarantees to specialized domains like code translation where reference validity is paramount.

## 6 CONCLUSION

We present Reference-Anchored Elo Estimation (RAEE), a framework that transforms simple pairwise comparisons into absolute, reproducible scores alongside uncertainty calibration. By anchoring all evaluations to a fixed reference and screening judges for reliability, RAEE mitigates scale drift, reduces dispersion across judges, and produces stable rankings even under changes in references used. Our theoretical analysis and empirical results confirm that RAEE preserves the discriminative power of pairwise judgments while achieving the interpretability and reproducibility that direct scoring has lacked.

As LLM judges become prevalent in evaluation pipelines, it is crucial that principled methods are used to ensure their reliability. RAEE provides a flexible architecture where any sufficiently consistent model can serve as a qualified judge, enabling scalable evaluation that adapts alongside the rapid progress of LLMs. Looking ahead, several directions remain open: detecting and correcting prompt-specific biases in judges, integrating RAEE with selective evaluation to balance reliability and efficiency, and extending the framework to broader domains where qualitative assessment requires stable absolute scales. We hope our work becomes a stepping stone for these future advances, and more broadly, toward rigorous and transferable LLM-based evaluation in related fields.

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

## A  LLM Usage Disclaimers

This paper was augmented in part with artificial intelligence tools during the preparation of the manuscript. Specifically, we used GPT-4.1 (OpenAI, 2025a) to assist with writing quality by improving clarity and readability (e.g., finding inconsistent notation usage, grammatical errors, etc.). Additionally, we used o3 (OpenAI, 2025b), alongside the `gpt-researcher` (Elovic, 2024) agent framework for literature exploration. These models were used solely as assistive tools in the drafting and research-gathering phases. All technical content, analysis, interpretations, and final conclusions are the authors' responsibility.

## B  Proofs and Technical Details for Section 2.3

### B.1  Setup and notation

For each prompt $i \in \{1, \ldots, N\}$, judge $j \in \{1, \ldots, J\}$, and run $r \in \{1, \ldots, k\}$, we observe an outcome

$$S_{ijr} \in \{1, \tfrac{1}{2}, 0\},$$

which encodes the result of model $B$ versus reference $A$ under RAEE (Section 2.1): $S_{ijr} = 1$ for a win of $B$, $S_{ijr} = 0$ for a loss, and $S_{ijr} = \tfrac{1}{2}$ for a tie. This ternary variable is treated as a bounded real number on $[0, 1]$.

**Two-way mixed ANOVA model.** To characterize reliability across repeated runs, we follow the classical Shrout–Fleiss ICC$(3, \cdot)$ framework (Shrout & Fleiss, 1979). The per-outcome decomposition is

$$S_{ijr} = \mu + \xi_i + \beta_j + \varepsilon_{ijr}. \tag{10}$$

Here: $\mu \in \mathbb{R}$ is the overall mean win probability (a fixed intercept). $\xi_i$ is a random effect capturing prompt-to-prompt difficulty variation. We assume $\mathbb{E}[\xi_i] = 0$ and $\mathrm{Var}(\xi_i) = \sigma_{\mathrm{p}}^2$, with $(\xi_i)$ independent and identically distributed (i.i.d.). $\beta_j$ is the fixed effect of judge $j$. Under the *consistency* ICC formulation, these fixed offsets do not contribute to covariance terms, but they allow different judges to have different baselines. $\varepsilon_{ijr}$ is the residual noise for judge $j$ on prompt $i$ in run $r$, assumed independent across $(i, j, r)$ with $\mathbb{E}[\varepsilon_{ijr}] = 0$ and $\mathrm{Var}(\varepsilon_{ijr}) = \sigma_{\mathrm{e}}^2$.

By construction, $\xi_i$ is independent of all $\varepsilon_{ijr}$. Thus, variance in observed scores arises from two sources: (i) systematic prompt variation $\sigma_{\mathrm{p}}^2$ and (ii) idiosyncratic run-level noise $\sigma_{\mathrm{e}}^2$.

The random prompt effect $\xi_i$ corresponds to the irreducible variability across different evaluation prompts. The residual $\varepsilon_{ijr}$ corresponds to instability in repeated evaluations of the same judge-prompt pair (e.g., due to nondeterminism or stochastic decoding). The fixed effect $\beta_j$ accounts for judge-specific calibration but cancels in correlations because ICC$(3,\cdot)$ uses a *consistency* definition rather than an *absolute agreement* definition.

This setup underlies the reliability analysis in Section 2.3, where we screen for ICC$(3, k)$ above a fixed threshold $\rho_0$ (typically $\rho_0 = 0.90$) to identify *strong judges*.

**Run and dataset means.**

$$\bar{S}_{ij\cdot} := \frac{1}{k} \sum_{r=1}^{k} S_{ijr}, \qquad \bar{S}_j^{(k)} := \frac{1}{N} \sum_{i=1}^{N} \bar{S}_{ij\cdot}. \tag{11}$$

$\bar{S}_{ij\cdot}$ is the average score for judge $j$ on prompt $i$ over $k$ runs, and $\bar{S}_j^{(k)}$ is the average score for judge $j$ over all $N$ prompts (after run averaging).

**Reliability.**

$$\rho_k := \mathrm{ICC}(3, k) = \frac{\sigma_{\mathrm{p}}^2}{\sigma_{\mathrm{p}}^2 + \sigma_{\mathrm{e}}^2/k} = \frac{k\,\rho_1}{1 + (k-1)\rho_1}, \qquad \rho_1 := \frac{\sigma_{\mathrm{p}}^2}{\sigma_{\mathrm{p}}^2 + \sigma_{\mathrm{e}}^2}. \tag{12}$$

This is the classical ICC(3,k) formula for the correlation of per-judge means $\bar{S}_j^{(k)}$ and $\bar{S}_{j'}^{(k)}$ over $N$ prompts, as shown in Proposition 3.

The final RAEE estimator is calculated as follows. Let $W, T, L$ be wins, ties, losses over the evaluation set (after any run-averaging). Set $S := W + \frac{1}{2}T$, $\hat{p} := S/N$, and Jeffreys-smoothed $\bar{p} := (S + \frac{1}{2})/(N + 1)$. Define Elo gap $\hat{\Delta}r := \gamma\,g(\bar{p})$ with $\gamma = 400/\ln 10$ (Algorithm 1).

### B.2 MOMENTS AND EXACT BETWEEN-JUDGE CORRELATION

**Lemma 2** (Moments of per-judge means). *Under equation 10–equation 11,*

$$\mathbb{E}[\bar{S}_j^{(k)}] = \mu + \beta_j, \quad \mathrm{Var}(\bar{S}_j^{(k)}) = \frac{\sigma_{\mathrm{p}}^2 + \sigma_{\mathrm{e}}^2/k}{N}, \quad \mathrm{Cov}(\bar{S}_j^{(k)}, \bar{S}_{j'}^{(k)}) = \frac{\sigma_{\mathrm{p}}^2}{N}\;(j \neq j').$$

Proof. *Shared $\xi_i$ drives the cross-judge covariance; runs average $\varepsilon_{ijr}$ to variance $\sigma_{\mathrm{e}}^2/k$.* □

**Proposition 3** (Correlation equals ICC$(3, k)$). *For $j \neq j'$,*

$$\mathrm{Corr}(\bar{S}_j^{(k)}, \bar{S}_{j'}^{(k)}) = \frac{\sigma_{\mathrm{p}}^2}{\sigma_{\mathrm{p}}^2 + \sigma_{\mathrm{e}}^2/k} = \rho_k.$$

Proof. *Divide the covariance and geometric mean of variances from Lemma 2; $N$ cancels.* □

**Remark 1** (Heteroscedastic runs). *If $\mathrm{Var}(\varepsilon_{ijr}) = \sigma_{\mathrm{e},j}^2$ depends on $j$, then $\mathrm{Corr}(\bar{S}_j^{(k)}, \bar{S}_{j'}^{(k)}) = \sqrt{\rho_{k,j}\rho_{k,j'}}$ with $\rho_{k,j} := \sigma_{\mathrm{p}}^2/(\sigma_{\mathrm{p}}^2 + \sigma_{\mathrm{e},j}^2/k)$. Screening at $\rho_{k,j} \geq \rho_0$ gives $\mathrm{Corr} \geq \rho_0$.*

### B.3 JEFFREYS SMOOTHING IS NEGLIGIBLE TO FIRST ORDER

**Lemma 3** (Shift bound).

$$\bar{p} - \hat{p} = \frac{N/2 - S}{N(N + 1)}, \qquad |\bar{p} - \hat{p}| \leq \frac{1}{2N}.$$

Proof. *Algebra on $(S + \frac{1}{2})/(N + 1) - S/N$. Use $0 \leq S \leq N$.* □

*Replacing $\hat{p}$ by $\bar{p}$ changes variances and covariances by $O(N^{-2})$ and correlations by $O(N^{-1})$. All $N^{-1}$ rates are unchanged.*

### B.4 TRANSPORT THROUGH THE ELO MAP

Let $g(p) = \gamma\,\mathrm{link}(p)$ and $\Delta r^{(j,k)} := g(\bar{S}_j^{(k)})$.

**Lemma 4** (Delta method). *Fix $p_\star := \mu \in (0, 1)$. Then $g'(p_\star) = \gamma/[p_\star(1 - p_\star)] > 0$ and*

$$\mathrm{Var}(\Delta r^{(j,k)}) = g'(p_\star)^2\,\mathrm{Var}(\bar{S}_j^{(k)}) + O(N^{-1})$$

$$\mathrm{Cov}(\Delta r^{(j,k)}, \Delta r^{(j',k)}) = g'(p_\star)^2\,\mathrm{Cov}(\bar{S}_j^{(k)}, \bar{S}_{j'}^{(k)}) + O(N^{-1}).$$

Proof. *First-order Taylor with bounded third moments; $\mathrm{Var}(\bar{S}_j^{(k)}) = \Theta(N^{-1})$.* □

**Proposition 4** (Correlation of Elo gaps).

$$\mathrm{Corr}(\Delta r^{(j,k)}, \Delta r^{(j',k)}) = \rho_k + O(N^{-1}).$$

Proof. *Apply Lemma 4 and Proposition 3; $g'(p_\star)^2$ cancels.* □

### B.5 POOLING ACROSS STRONG JUDGES

**Proposition 5** (Variance under judge pooling). *Let $\widehat{\Delta r}^{(k)} := J^{-1}\sum_{j=1}^{J}\Delta r^{(j,k)}$ and denote $\sigma_{\Delta,k}^2 := \mathrm{Var}(\Delta r^{(j,k)})$ (identical up to $O(N^{-1})$). Then*

$$\mathrm{Var}(\widehat{\Delta r}^{(k)}) = \sigma_{\Delta,k}^2\Big(\rho_k + \frac{1 - \rho_k}{J}\Big) + O(N^{-1}).$$

Proof. *With exchangeable covariance, $\mathrm{Cov}(\Delta r^{(j,k)}, \Delta r^{(j',k)}) = \rho_k\sigma_{\Delta,k}^2 + O(N^{-1})$ for $j \neq j'$. Compute the variance of the mean.* □

**Rate form.**    Using Lemma 4 and Lemma 2,

$$\text{Var}(\widehat{\Delta r}^{(k)}) = \frac{\gamma^2}{\mu^2(1-\mu)^2} \cdot \frac{\sigma_{\text{p}}^2 + \sigma_{\text{e}}^2/k}{N} \cdot \left(\rho_k + \frac{1-\rho_k}{J}\right) + O(N^{-1}).$$

Only the idiosyncratic $(1 - \rho_k)$ part shrinks like $J^{-1}$; the shared prompt component $\rho_k$ does not.

## B.6    MAP BACK TO WIN PROBABILITY

Let $h(x) = \sigma\left(\frac{\ln 10}{400} x\right)$ and $\widehat{P}^{(k)} := h(\widehat{\Delta r}^{(k)})$.

**Lemma 5** (Delta through $h$).    *With $x_\star = g(\mu)$,*

$$\text{Var}\big(h(\Delta r^{(j,k)})\big) = h'(x_\star)^2 \, \text{Var}(\Delta r^{(j,k)}) + O(N^{-1}),$$

$$\text{Cov}\big(h(\Delta r^{(j,k)}), h(\Delta r^{(j',k)})\big) = h'(x_\star)^2 \, \text{Cov}(\Delta r^{(j,k)}, \Delta r^{(j',k)}) + O(N^{-1}).$$

Proof. *First-order Taylor at $x_\star$.*                                                                    □

**Corollary 2** (Stable win probability)**.**

$$\text{Corr}\big(h(\Delta r^{(j,k)}), h(\Delta r^{(j',k)})\big) = \rho_k + O(N^{-1}), \quad \text{Var}\big(h(\widehat{\Delta r}^{(k)})\big) = h'(x_\star)^2 \, \text{Var}(\widehat{\Delta r}^{(k)}) + O(N^{-1}).$$

## B.7    UNCERTAINTY QUANTIFICATION AND FINITE-POPULATION CORRECTION

**Posterior and exact Elo interval.**    With Jeffreys prior $\text{Beta}(\frac{1}{2}, \frac{1}{2})$ and the composite Bernoulli log-likelihood on $S_i \in \{0, \frac{1}{2}, 1\}$,

$$p \mid \{S_i\} \sim \text{Beta}(a, b), \quad a = W + \tfrac{1}{2}T + \tfrac{1}{2}, \quad b = L + \tfrac{1}{2}T + \tfrac{1}{2}, \quad \bar{p} = \frac{a}{a+b} = \frac{S + \frac{1}{2}}{N+1}.$$

The $(1 - \alpha)$ credible interval is $[p_\ell, p_u] = [\text{BetaQ}(\alpha/2; a, b), \text{BetaQ}(1 - \alpha/2; a, b)]$ and the Elo interval is the monotone image

$$[\Delta r_\ell, \Delta r_u] = \gamma \, [g(p_\ell), g(p_u)].$$

**Delta sanity check.**    Using $\text{Var}_{\text{beta}}(p) = ab/[(a + b)^2(a + b + 1)]$ and $g'(p) = \gamma/[p(1 - p)]$,

$$\text{SE}(\hat{\Delta}r) \approx \frac{\gamma}{\bar{p}(1 - \bar{p})} \sqrt{\frac{ab}{(a + b)^2(a + b + 1)}}.$$

This matches the delta-method SE from Lemma 2 and Lemma 4 to first order in $N^{-1}$, while also showing $\Theta(N^{-1/2})$ decay.

## B.8    ANCHOR ADDITIVITY UNDER A DIFFERENCE MODEL

Assume there exist latent $\{\lambda_m\}$ and a strictly increasing $T : (0, 1) \to \mathbb{R}$ such that $T(p_{ij}) = \lambda_i - \lambda_j$. Then for any $A, B, C$:

$$\Delta r_{BC} = \kappa T(p_{BC}) = \kappa T(p_{BA}) + \kappa T(p_{AC}) = \Delta r_{BA} + \Delta r_{AC}.$$

This requires only a difference structure and monotonicity, not a specific tie model.

## B.9    BOUNDARIES AND CAVEATS

**Boundaries.**    $g'(p)$ and $h'$ blow up near $p \in \{0, 1\}$. Jeffreys smoothing keeps $\bar{p} \in (0, 1)$ and stabilizes the delta approximations provided $\bar{p} \in [\epsilon, 1 - \epsilon]$ for fixed $\epsilon > 0$.

**Judge fixed effects.**    Consistency $\text{ICC}(3, \cdot)$ removes $\beta_j$ from covariance structure; results depend only on $(\sigma_{\text{p}}^2, \sigma_{\text{e}}^2)$.

## B.10 SUMMARY AND IMPLICATIONS

In this section we discussed the reliability of the RAEE framework. The main finding is that the run-averaged per-judge means exhibit exact correlation $\rho_k$, and this correlation transfers to Elo gaps up to $O(N^{-1})$ error, ensuring that strong judges remain highly concordant.

When pooling across $J$ judges, the variance of the aggregated Elo score decomposes into an irreducible prompt component and a reducible idiosyncratic component, giving $\mathrm{Var}(\widehat{\Delta r}^{(k)}) = \sigma_{\Delta,k}^2(\rho_k + (1 - \rho_k)/J) + O(N^{-1})$ with $\sigma_{\Delta,k}^2$ scaling as $1/N$. The logistic link from win probability to Elo preserves both the correlation structure and the $N^{-1/2}$ convergence rate. For single-run evaluations, the effective reliability adjusts to $\rho_{\mathrm{EVAL}} = \rho_k/[k - (k-1)\rho_k]$.

Finally, uncertainty quantification is provided by Jeffreys-Beta posterior intervals on the win probability, whose monotone images yield valid Elo intervals, and finite-population correction $(M - n)/(M - 1)$ applies when prompts are sampled without replacement.

# C DETAILED TABLES AND RESULTS FOR SECTION 3

This appendix supports the main claims with compact evidence. We first show why direct scoring is unstable, then examine judge reliability, validate RAEE's analytic uncertainty (including finite-population effects), and finally test cross-domain robustness.

## C.1 DIRECT SCORING RESULTS: EVIDENCE OF SYSTEMATIC INSTABILITIES

**Setup.** Twenty judge models score three targets (*gemini-2.5-flash*, *gpt-4o*, *o4-mini*) on MT-Bench. Each judge runs five times, two turns, 80 prompts (800 judgments per judge-target). All judges use the same 1-10 direct scoring prompt.

Table 8: **Direct scoring pathologies.** Per judge-target: mean, SD, kurtosis from 800 judgments. High means with large kurtosis indicate ceiling compression; between-judge mean spread shows scale drift.

| Judge | *gemini-2.5-flash* | | | *gpt-4o* | | | *o4-mini* | | |
|---|---|---|---|---|---|---|---|---|---|
| | Mean | SD | Kurt. | Mean | SD | Kurt. | Mean | SD | Kurt. |
| claude-opus-4 | 8.326 | 1.825 | 5.058 | 7.563 | 2.031 | 0.273 | 8.026 | 1.941 | 2.950 |
| claude-sonnet-4 | 8.317 | 1.827 | 4.829 | 7.551 | 1.999 | 0.273 | 8.029 | 1.881 | 3.074 |
| DeepSeek-V3-0324 | 8.617 | 1.891 | 6.308 | 8.511 | 1.482 | 4.644 | 8.399 | 1.908 | 5.052 |
| gemini-2.0-flash | 9.331 | 1.924 | 10.950 | 9.169 | 1.833 | 7.764 | 9.304 | 2.008 | 10.930 |
| gemini-2.0-flash-lite | 9.426 | 1.701 | 16.087 | 9.130 | 1.593 | 8.593 | 9.208 | 1.731 | 11.055 |
| gemini-2.5-flash | 9.618 | 1.382 | 26.046 | 8.764 | 2.140 | 2.079 | 9.516 | 1.639 | 17.243 |
| gemini-2.5-flash-lite-preview-06-17 | 9.296 | 1.615 | 15.182 | 8.393 | 2.324 | 2.424 | 9.018 | 1.795 | 8.991 |
| gemini-2.5-pro | 9.605 | 1.408 | 19.459 | 8.673 | 2.446 | 1.278 | 9.393 | 1.842 | 9.918 |
| gpt-4.1 | 9.278 | 1.726 | 11.286 | 8.914 | 1.478 | 4.952 | 8.889 | 2.026 | 4.558 |
| gpt-4o | 8.744 | 2.045 | 5.424 | 8.665 | 1.685 | 4.760 | 8.169 | 2.300 | 1.558 |
| Llama-3.3-70B | 8.712 | 1.720 | 11.628 | 8.536 | 1.628 | 10.103 | 8.491 | 2.021 | 6.711 |
| Llama-4-Scout | 8.808 | 1.506 | 15.471 | 8.666 | 1.400 | 9.435 | 8.478 | 1.763 | 7.999 |
| o3 | 8.024 | 2.204 | 2.667 | 7.379 | 2.294 | −0.109 | 7.958 | 2.167 | 3.256 |
| o3-mini | 9.211 | 1.939 | 8.754 | 8.938 | 1.780 | 3.771 | 9.080 | 2.211 | 6.230 |
| o4-mini | 8.659 | 2.081 | 4.882 | 7.928 | 2.355 | 1.001 | 8.863 | 1.817 | 8.535 |
| Qwen3-235B-A22B | 9.375 | 1.099 | 12.880 | 8.643 | 1.717 | 2.645 | 9.006 | 1.649 | 9.357 |
| Qwen3-235B-A22B-Thinking-2507 | 9.375 | 1.227 | 14.266 | 8.448 | 2.020 | 1.269 | 9.368 | 1.365 | 17.770 |
| Qwen3-30B-A3B | 9.763 | 1.193 | 29.904 | 9.091 | 2.086 | 3.709 | 9.415 | 1.797 | 9.421 |
| Qwen3-32B | 9.300 | 1.228 | 13.231 | 8.643 | 1.746 | 2.713 | 9.090 | 1.614 | 10.895 |
| Qwen3-Coder | 8.554 | 1.576 | 3.900 | 7.606 | 2.023 | −0.089 | 8.099 | 1.889 | 2.518 |
| **Total** | 9.017 | 1.656 | 11.911 | 8.461 | 1.903 | 3.574 | 8.790 | 1.868 | 7.901 |

**Takeaways.** (i) Scores compress near 8-10 with heavy tails (kurtosis $> 10$ for many judges). (ii) Means vary by up to $\sim 2$ points across judges for the same target (scale drift). (iii) Discrimination is inconsistent across judges. These motivate a reference-anchored approach.

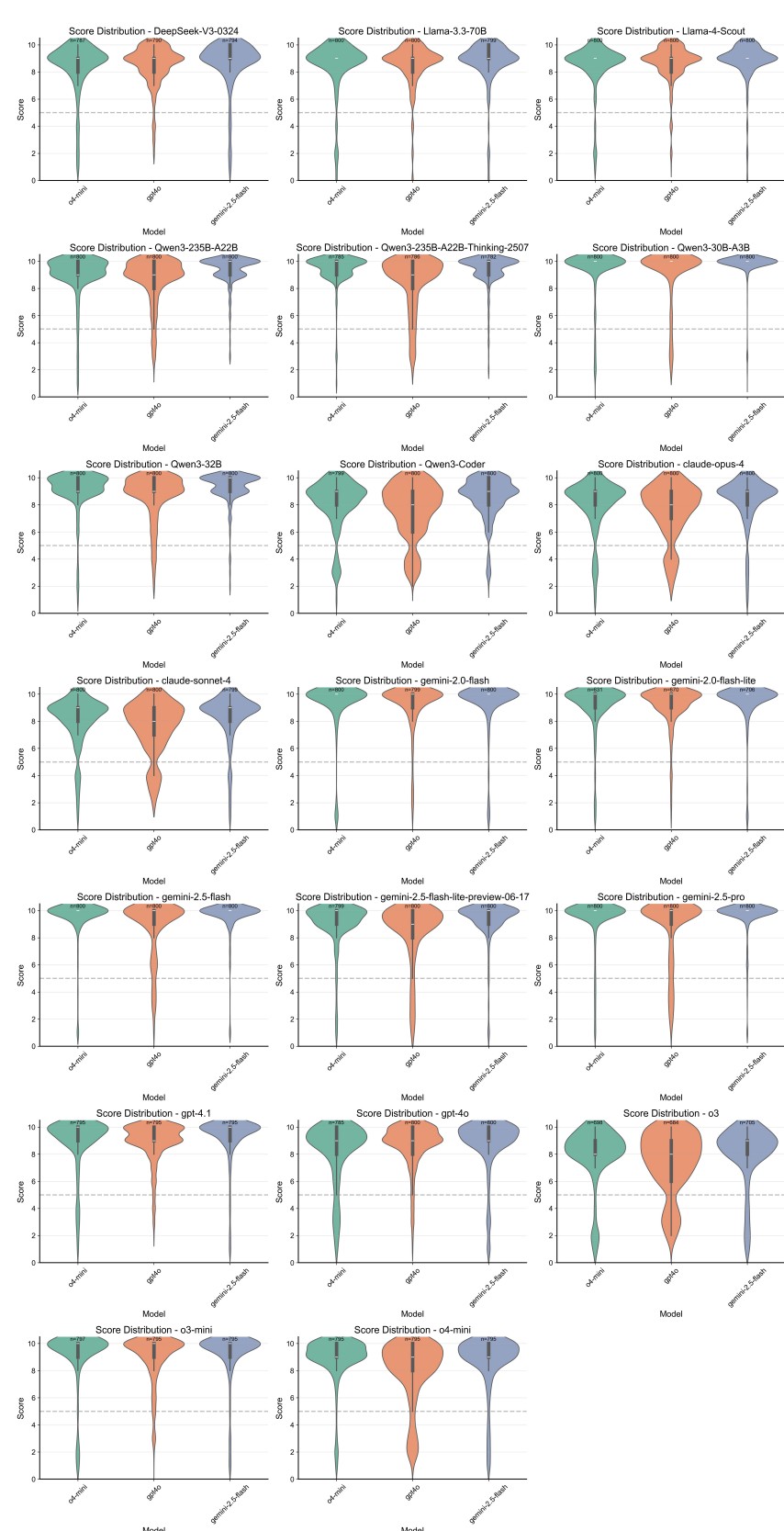

Figure 6: **Direct scoring distributions.** Violin plots by judge and target. Mass concentrates at 8-10; shapes differ across judges. Small perturbations can flip rankings.

Table 9: **SE vs. subsample size** $n$**.** Empirical bootstrap vs. analytic SE. The $\kappa$ adjustment accounts for judge reliability.

| $n$ | Emp. SE | Theor. SE | FPC-corr. SE | $\kappa$-adj. SE | Emp/Theor | Emp/Adj | Emp/$\kappa$ |
|---|---|---|---|---|---|---|---|
| 16 | 0.085656 | 0.117833 | 0.112137 | 0.095316 | 0.726925 | 0.763848 | 0.898644 |
| 24 | 0.072636 | 0.098042 | 0.090674 | 0.077073 | 0.740862 | 0.801062 | 0.942426 |
| 32 | 0.062976 | 0.085735 | 0.076925 | 0.065386 | 0.734537 | 0.818667 | 0.963137 |
| 40 | 0.056197 | 0.077139 | 0.067014 | 0.056962 | 0.728524 | 0.838594 | 0.986581 |
| 64 | 0.039337 | 0.061535 | 0.047815 | 0.040642 | 0.639253 | 0.822689 | 0.967869 |
| 80 | 0.030263 | 0.055206 | 0.039159 | 0.033285 | 0.548177 | 0.772813 | 0.909192 |
| 120 | 0.018033 | 0.045260 | 0.022701 | 0.019296 | 0.398423 | 0.794352 | 0.934531 |

Table 10: **FloRES translation.** Win probabilities $\hat{p}$ and Elo gaps vs. human references across four languages. Rankings are stable; SEs shrink with strong judges.

| Model | Lang. | Mean $\hat{p}$ | | Mean $\Delta r$ | | CV($\hat{p}$) | Strong judges | |
|---|---|---|---|---|---|---|---|---|
| | | mean | SE | mean | SE | | SE($\hat{p}$) | SE($\Delta r$) |
| *gemini-2.5-flash* | *zh* | 0.2761 | 0.0287 | -189.79 | 5.01 | 0.1433 | 0.0266 | 4.65 |
| | *ja* | 0.2618 | 0.0278 | -208.50 | 4.85 | 0.1847 | 0.0255 | 4.46 |
| | *ko* | 0.2539 | 0.0281 | -212.33 | 4.91 | 0.1382 | 0.0258 | 4.51 |
| | *es* | 0.3813 | 0.0312 | -91.01 | 5.44 | 0.0728 | 0.0310 | 5.41 |
| *gpt4o* | *zh* | 0.2958 | 0.0284 | -182.56 | 4.96 | 0.0899 | 0.0252 | 4.40 |
| | *ja* | **0.2885** | 0.0287 | -167.96 | 5.02 | 0.0817 | 0.0259 | 4.51 |
| | *ko* | **0.2881** | 0.0291 | -180.74 | 5.08 | 0.1205 | 0.0262 | 4.57 |
| | *es* | 0.3811 | 0.0284 | -103.70 | 4.95 | 0.1202 | 0.0274 | 4.79 |
| *o4-mini* | *zh* | **0.2966** | 0.0294 | -167.69 | 5.14 | 0.1053 | 0.0278 | 4.85 |
| | *ja* | 0.2719 | 0.0283 | -197.79 | 4.93 | 0.1171 | 0.0261 | 4.56 |
| | *ko* | 0.2558 | 0.0282 | -208.40 | 4.93 | 0.1013 | 0.0259 | 4.53 |
| | *es* | **0.3959** | 0.0317 | -78.72 | 5.54 | 0.0942 | 0.0317 | 5.53 |

## C.2 FINITE-POPULATION EFFECTS: VALIDATING THEORETICAL UNCERTAINTY

We compare empirical SEs from bootstrap to RAEE's analytic SE with finite-population correction (FPC) for $M = 160$ prompts and a reliability adjustment $\kappa$.

Takeaways. Uncorrected theory is conservative. FPC aligns analytic SEs with bootstrap. A simple reliability factor ($\kappa$) further improves calibration. RAEE yields closed-form, accurate intervals without resampling.

**Reliability adjustment** $\kappa$**.** The factor $\kappa \approx 0.85$ is an empirically-derived constant that accounts for inter-judge reliability. It is computed as the square root of the average Cohen's $\kappa$ across all judge pairs, measuring agreement beyond chance. The $\kappa$-adjusted SE is calculated as: Analytic SE $\times$ FPC $\times \sqrt{\kappa}$, where the square root ensures proper variance scaling. This adjustment brings theoretical predictions within $\pm 12\%$ of empirical bootstrap estimates across all sample sizes (see Appendix E.4 for calibration details on new domains).

## C.3 CROSS-DOMAIN ROBUSTNESS

We apply RAEE to translation (FloRES) and summarization (TL;DR). Results are summarized below; "strong judges" are those passing ICC screening.

Takeaways. Rankings are stable across languages and categories. Translation sits below human references; summarization often exceeds them. ICC screening tightens intervals.

Table 11: **TL;DR summarization.** Models vs. human summaries across four categories. High $\hat{p}$ (0.78-0.96). Rankings are consistent; TIFU is hardest.

| Model | Cat. | Mean $\hat{p}$ | | Mean $\Delta r$ | | CV($\hat{p}$) | Strong judges | |
|---|---|---|---|---|---|---|---|---|
| | | mean | SE | mean | SE | | SE($\hat{p}$) | SE($\Delta r$) |
| *gemini-2.5-flash* | *Misc.* | 0.9614 | 0.0146 | 601.14 | 2.54 | 0.0132 | 0.0129 | 2.24 |
| | *AskReddit* | 0.9607 | 0.0148 | 594.03 | 2.58 | 0.0107 | 0.0130 | 2.28 |
| | *Relationships* | 0.9515 | 0.0163 | 559.47 | 2.85 | 0.0159 | 0.0143 | 2.50 |
| | *TIFU* | 0.9216 | 0.0192 | 459.05 | 3.35 | 0.0197 | 0.0180 | 3.14 |
| *gpt4o* | *Misc.* | 0.9552 | 0.0153 | 557.61 | 2.67 | 0.0147 | 0.0403 | 2.45 |
| | *AskReddit* | 0.9538 | 0.0152 | 549.19 | 2.65 | 0.0163 | 0.0142 | 2.49 |
| | *Relationships* | 0.9471 | 0.0165 | 531.45 | 2.89 | 0.0171 | 0.0151 | 2.64 |
| | *TIFU* | 0.8797 | 0.0225 | 358.42 | 3.93 | 0.0186 | 0.0223 | 3.90 |
| *o4-mini* | *Misc.* | 0.9341 | 0.0176 | 467.57 | 3.07 | 0.0194 | 0.0173 | 3.01 |
| | *AskReddit* | 0.9120 | 0.0200 | 409.47 | 3.49 | 0.0184 | 0.0198 | 3.46 |
| | *Relationships* | 0.9254 | 0.0186 | 444.24 | 3.25 | 0.0200 | 0.0183 | 3.19 |
| | *TIFU* | 0.7833 | 0.0290 | 225.04 | 5.07 | 0.0039 | 0.0288 | 5.04 |

## C.4 SYNTHESIS: APPENDIX FINDINGS AND IMPLICATIONS

Direct scoring shows ceiling compression, scale drift, and unstable rankings (Table 8, Fig. 6). RAEE resolves these via reference anchoring and delivers empirically conservative uncertainty with simple corrections (Table 9). Results generalize across translation and summarization with consistent rankings and interpretable task difficulty (Tables 10-11). The evidence supports RAEE as a practical default for LLM evaluation.

## D JUDGE PROMPTS

This section provides the specific prompts used by LLM judges in our RAEE evaluation framework. The prompts are designed for pairwise comparison between a system response and a reference response, returning ternary outcomes (win/tie/loss) that feed into the reference-anchored Elo estimation process described in Section 2. These standardized prompts ensure consistent evaluation across different judge models while maintaining the reliability requirements outlined in Section 2.3.

## E SENSITIVITY AND ROBUSTNESS ANALYSIS

This appendix provides detailed evidence for RAEE's robustness across different evaluation conditions, including abstention filtering, smoothing prior choices, reliability adjustments, and conceptual comparisons with related benchmarks.

### E.1 ANCHOR SELECTION AND DIAGNOSTICS

The validity of RAEE relies on the reference anchor providing a meaningful baseline for comparison. An overly weak or strong anchor can compress win probabilities near 0 or 1, reducing discriminative power. To ensure anchor appropriateness, we recommend two key diagnostics:

- **Mid-region Coverage Diagnostic:** We flag anchors that exhibit excessive floor or ceiling effects. A suitable anchor should have at least 95% of its win probabilities $\bar{p}$ within the informative range $[0.2, 0.8]$. Anchors violating this (e.g., $\bar{p} > 0.95$ for almost all prompts) should be rejected as too weak or too strong.

- **Anchor-Swap Invariance:** To verify that the choice of anchor does not distort rankings, we compute Kendall's $\tau$ rank correlation between leaderboards generated by different candidate anchors. We also check that the centered Elo shifts remain stable (low variance in $\Delta r$ shifts).

We recommend selecting anchors that maximize mid-region coverage while maintaining high cross-anchor rank correlations ($\tau > 0.9$).

### E.2 IMPACT OF ABSTENTION (TRUST-OR-ESCALATE)

We evaluate RAEE's stability when judges abstain on uncertain prompts using a "Trust-or-Escalate" filter. We implement an abstention policy where the system refuses to score a prompt if the across-run variance exceeds a fixed threshold (std > 0.5 on 1-10 scale, or equivalent win-probability variance). Table 6 shows that RAEE maintains low coefficient of variation (CV) even when filtering is applied, outperforming direct scoring in both filtered and unfiltered settings.

We also verify that abstention does not distort the underlying preference structure. Table 12 shows that even when 5–12% of prompts are filtered out, the shifts in win probability ($\Delta\bar{p}$) and Elo ($\Delta$Elo) are minimal, and the relative ranking of models remains stable.

Table 12: **Stability of RAEE Scores under Abstention.** Filtering noisy prompts results in minimal shifts in win probability and Elo scores, indicating that RAEE's rankings are robust to the inclusion of uncertain items.

| Ref. | Model | Cov. | $\bar{p}_{\text{all}}$ | $\bar{p}_{\text{trust}}$ | $\Delta\bar{p}$ | $\text{Elo}_{\text{all}}$ | $\text{Elo}_{\text{trust}}$ | $\Delta$Elo |
|------|-------|------|------|------|------|------|------|------|
| O4-Mini | GPT-4o | 0.9500 | 0.3497 | 0.3382 | −0.0116 | −107.74 | −116.66 | −8.92 |
| O4-Mini | Llama-4 | 0.9563 | 0.3082 | 0.2983 | −0.0099 | −140.48 | −148.60 | −8.12 |
| GPT-4o | Gemini-F | 0.8812 | 0.5079 | 0.5074 | −0.0005 | 5.51 | 5.16 | −0.35 |
| GPT-4o | Qwen-A22B | 0.8875 | 0.4321 | 0.4253 | −0.0067 | −47.50 | −52.28 | −4.78 |

**Takeaway:** RAEE's coefficient of variation remains low ($< 0.10$) regardless of abstention filtering, while direct scoring shows high variability (CV $> 0.26$) even after filtering (Table 6). Furthermore, RAEE can be effectively combined with judge-level abstention (e.g., Trust-or-Escalate) to further refine reliability, though the gains are modest because RAEE already suppresses most reducible variance.

### E.3 SMOOTHING SENSITIVITY (JEFFREYS VS. LAPLACE)

We test the sensitivity of RAEE to the choice of prior by comparing Jeffreys smoothing ($\alpha = \beta = 0.5$) with Laplace smoothing ($\alpha = \beta = 1$). Table 13 shows that the choice of prior has negligible impact on rankings and Elo estimates.

Table 13: **Smoothing Sensitivity (Jeffreys vs. Laplace).** Rankings and Elo estimates are nearly identical regardless of smoothing prior choice.

| Anchor | Kendall $\tau$ | MAE (Elo) | Max Diff (Elo) | % within $\pm20$ Elo |
|--------|------|------|------|------|
| FloRES | 1.00 | 0.10 | 0.10 | 100% |
| MT-Bench | 1.00 | 0.13 | 0.30 | 100% |
| TL;DR | 1.00 | 0.20 | 0.30 | 100% |

**Takeaway:** The choice between Jeffreys and Laplace priors is negligible in practice. Perfect rank correlation ($\tau = 1.0$) and minimal Elo differences ($< 0.3$ points) confirm that RAEE's results are robust to this modeling choice.

### E.4 RELIABILITY ADJUSTMENT CALIBRATION ($\kappa$)

The reliability adjustment factor $\kappa$ accounts for inter-judge agreement beyond the finite-population correction. Table 14 shows that the $\kappa$-adjusted analytic SE closely matches empirical bootstrap estimates across tasks.

**Takeaway:** The $\kappa$ adjustment (computed as $\sqrt{\text{mean Cohen's } \kappa}$ across judge pairs) provides accurate uncertainty estimates without resampling. For new domains (e.g., code generation, reasoning), one can calibrate this factor by running the judge pool on a small sample of prompts, computing the

Table 14: $\kappa$ **Reliability Adjustment Calibration.** The $\kappa$ factor brings analytic SEs within $\pm 4\%$ of empirical estimates.

| Task | Empirical SE | Analytic SE (Raw) | FPC Adjusted | $\kappa$-Adjusted | Emp / $\kappa$-Adj |
|---|---|---|---|---|---|
| MT-Bench | 0.0562 | 0.0771 | 0.0670 | 0.0570 | 0.99 |
| FloRES | 0.0058 | 0.0066 | 0.0062 | 0.0058 | 1.00 |
| TL;DR | 0.0223 | 0.0244 | 0.0232 | 0.0214 | 1.04 |

Table 15: **RAEE vs. Arena-Hard Auto Leaderboard Comparison (72 Models).** RAEE recovers the standard leaderboard on a dense field while adding uncertainty quantification.

| Metric | Value |
|---|---|
| Kendall $\tau$ (rank correlation) | 0.9945 |
| Spearman $\rho$ (rank correlation) | 0.9997 |
| MAD in Elo difference ($\Delta r$) | 8.52 |
| MAD in win-rate (percentage points) | 0.71 |
| Top-10 set overlap (Jaccard) | 1.00 |

average pairwise Cohen's $\kappa$, and setting the adjustment factor to $\sqrt{\kappa_{\text{Cohen}}}$. The ratio of empirical to $\kappa$-adjusted SE is within $\pm 4\%$ across all tasks.

To verify RAEE's performance in a dense, competitive setting, we evaluated it on the **Arena-Hard Auto** dataset containing 72 models with small capability gaps. Table 15 shows that RAEE faithfully reproduces the official leaderboard rankings (Kendall $\tau \approx 0.995$, Top-10 overlap 1.0) while providing the benefits of anchored probability interpretation.

### E.5 JUDGE MODEL DETAILS

Table 16 lists all judge models used in our experiments, categorized by their ICC screening status.

Table 16: **Judge Model Details.** Strong judges are those passing $\text{ICC}(3, k) \geq 0.90$ screening.

| Role | Models | Count |
|---|---|---|
| **Strong Judges** | Qwen3-30B-A3B, Qwen3-235B-A22B, o3-mini, gpt-4o, Qwen3-Coder, Llama-4-Scout, o3, Llama-3.3-70B, Qwen3-32B | 9 |
| **Other Judges** | claude-opus-4, claude-sonnet-4, DeepSeek-V3-0324, gemini-2.0-flash, gemini-2.0-flash-lite, gemini-2.5-flash, gemini-2.5-flash-lite-preview-06-17, gemini-2.5-pro, gpt-4.1, o4-mini, Qwen3-235B-A22B-Thinking-2507 | 11 |

### E.6 CONCEPTUAL COMPARISON WITH RELATED BENCHMARKS

Table 17 positions RAEE relative to Arena-Hard Auto and WildBench (WB-Reward), highlighting RAEE's unique contributions in uncertainty quantification and reliability screening.

**Takeaway:** RAEE is the only framework that combines anchored pairwise comparisons with closed-form uncertainty and ICC-based reliability guarantees. This makes it a natural "reliability layer" for existing benchmarks like Arena-Hard Auto and WildBench.

### E.7 STATISTICAL VS. COGNITIVE BIASES

It is important to distinguish between *statistical biases* and *cognitive biases* in LLM evaluation.

1) **Statistical Biases:** Issues such as scale drift, ceiling compression, and uncalibrated uncertainty are structural failures of the scoring mechanism. RAEE explicitly resolves these through reference anchoring (fixing the scale) and Jeffreys-Beta modeling (calibrating uncertainty).

2) **Cognitive Biases:** Issues such as the "comparative trap" (preferring fluent but incorrect answers), self-preference, and verbosity bias are intrinsic to the judge models themselves. RAEE does not

Table 17: **Conceptual Comparison.** RAEE provides the reliability layer (analytic uncertainty + ICC screening) that existing benchmarks lack.

| Feature | RAEE | Arena-Hard Auto | WB-Reward |
|---|---|---|---|
| **Comparison** | Anchored Pairwise | Tournament Pairwise | Anchored Pairwise |
| **Scale** | Portable Absolute ($\Delta r$) | Relative Elo | Relative Score |
| **Uncertainty** | **Analytic (Closed-form)** | Bootstrap | None |
| **Reliability** | **ICC Screening** | None | None |

inherently remove these biases. However, by providing rigorous uncertainty quantification and reliability screening, RAEE makes the impact of these biases visible (e.g., as high variance or low inter-judge agreement). Furthermore, RAEE's modular design allows it to be layered with judge-level interventions, such as abstention filters (Appendix E.2) or debiased prompting strategies, without altering the aggregation framework.

```
You are a helpful assistant.

[Instruction]
Please act as an impartial judge and evaluate
the quality of the response provided by an AI assistant
to the user question displayed below. Your evaluation
should consider correctness and helpfulness.
You will be given a reference answer and the assistant's answer.
Begin your evaluation by comparing the assistant's answer
with the reference answer. Identify and correct any mistakes.
Be as objective as possible.
After providing your explanation, you must rate the
response on a scale of 1 to 10 by strictly following this format:
\"[[rating]]\", for example: \"Rating: [[5]]\".","

[Question]
${question_1}

[The Start of Reference Answer]
${ref_answer_1}
[The End of Reference Answer]

[The Start of Assistant's Answer]
${answer_1}
[The End of Assistant's Answer]","{system_prompt}

<|The Start of Reference Answer|>

### User:
${question_1}

### Reference answer:
${ref_answer_1}

### User:
${question_2}

### Reference answer:
${ref_answer_2}

<|The End of Reference Answer|>

<|The Start of Assistant A's Conversation with User|>

### User:
${question_1}

### Assistant A:
${answer_1}

### User:
${question_2}

### Assistant A:
${answer_2}

<|The End of Assistant A's Conversation with User|>
```

Figure 7: Direct judge prompt (MT-Bench).

```
Please act as an impartial judge and evaluate the quality of the
responses provided by two AI assistants to the user question
displayed below. You should choose the assistant that follows the
user's instructions and answers the user's question better. Your
evaluation should consider factors such as helpfulness, relevance,
accuracy, depth, creativity, and level of detail of their responses.
Begin your evaluation by providing a short explanation. Avoid any
position biases and ensure that the order in which the responses
were presented does not affect your judgment. Do not allow the
length of the responses to influence your evaluation. Do not favor
certain names of the assistants. Be as objective as possible.
After providing your explanation, you must rate the response that
is better. Rate the response that is significantly better as "A"
or "B", and rate them as tied "C" if they are roughly equal.
Format your answer as a json object strictly following:
```json
{"score": "A" | "B" | "C",
 "confidence": 0.0 | 1.0,
 "reason": "Your reason for the score"}
```

[User Question]
{question_1}

[The Start of Assistant A's Answer]
{ref_answer_1}
[The End of Assistant A's Answer]

[The Start of Assistant B's Answer]
{answer_1}
[The End of Assistant B's Answer]
```

Figure 8: Pairwise judge prompt (MT-Bench).

```
Select the Output A or Output B that is better for the given instruction.

Here are some rules of the evaluation:
(1) You should prioritize evaluating whether the output
honestly/precisely/closely executes the instruction,
then consider its helpfulness, accuracy, level of detail,
harmlessness, etc.
(2) Outputs should NOT contain more/less than what the
instruction asks for, as such outputs do NOT precisely
execute the instruction.
(3) You should avoid any potential bias and your judgment
should be as objective as possible. For example, the order
in which the outputs were presented should NOT affect your
judgment, as Output (a) and Output (b) are **equally likely**
to be the better.

Here is the example of the instruction and outputs:
```
Instruction: Translate the following text into English.
The translation should be accurate and maintain the
original meaning. DO NOT output any other text.

Text: {question}

Output A: {reference}

Output B: {answer}
```

Do NOT provide any explanation for your choice.
Do NOT say both / neither are good.

Your final verdict by strictly following this format:
"A" if assistant A is better, "B" if assistant B is better,
and "C" for a tie. Format your answer as a json object
strictly following the following format:
```json
{
"score": "A" | "B" | "C",
"confidence": 0.0 | 1.0,
}
```

Do NOT output any other words.
```

Figure 9: Pairwise judge prompt (FloRES).

```
Select the Output A or Output B that is better for the given instruction.

Here are some rules of the evaluation:
(1) You should prioritize evaluating whether the output
honestly/precisely/closely executes the instruction,
then consider its helpfulness, accuracy, level of detail,
harmlessness, etc.
(2) Outputs should NOT contain more/less
than what the instruction asks for,
as such outputs do NOT precisely execute the instruction.
(3) You should avoid any potential bias and your judgment
should be as objective as possible. For example,
the order in which the outputs were presented should NOT
affect your judgment, as Output (a) and Output (b) are
**equally likely** to be the better.

Here is the example of the instruction and outputs:
```
Instruction: Write a concise TL;DR of the following text in
first-person POV. The summary should be no longer than 30 words
and should capture the main points of the text without losing
important details.

Text: {question}

Output A: {reference}

Output B: {answer}
```

Do NOT provide any explanation for your choice.
Do NOT say both / neither are good.

Your final verdict by strictly following this format:
"A" if assistant A is better, "B" if assistant B is better,
and "C" for a tie. Format your answer as a json object
strictly following the following format:
```json
{
"score": "A" | "B" | "C",
"confidence": 0.0 | 1.0,
}
```

Do NOT output any other words.
```

Figure 10: Pairwise judge prompt (TL;DR).

