# OpenReview forum: "Robust LLM-Based Scoring via Reference-Anchored ELO Estimation"
_ICLR.cc/2026/Conference — Submitted to ICLR 2026_

### Official Review · Reviewer_sUeC · 2025-10-15

**Soundness:** 2
**Presentation:** 3
**Contribution:** 1
**Rating:** 2
**Confidence:** 4

**Summary:**

The authors present Reference-Anchored Elo-Estimation (RAEE), a principled framework that anchors all model comparisons
to a fixed reference and expresses outcomes as win probabilities on a relative scale as an alternative to absolute scoring. The authors compare to absolute scoring and find that RAEE reduces per-run standard error by ≈ 44% and across-judge coefficient of variation by ≈ 72% relative to direct scoring, while preserving ranking stability even under reference changes.

**Strengths:**

* The paper's presentation and styling are strong.
* The paper is reasonably well organized.

**Weaknesses:**

* The main contribution of this work is mostly anticipated by Arena-Hard Auto (https://arxiv.org/abs/2406.11939), an extremely popular LLM-judged alignment benchmark from the creators of Chatbot Arena published in 2024, making this a largely derivative work unsuitable for publication.
* Even were this not the case, ELO scoring introduces its own set of biases, including the enforcement of transitive preference where, in many cases, none exists, and the hyperparameter choice of baseline model for comparison, which can be highly biasing (https://arxiv.org/abs/2509.20293, https://arxiv.org/abs/2409.15268). The authors' mechanism is largely standard and does not remediate this.
* The paper cites some highly relevant related works, e.g. Khurana et al., 2024, Zhang et al., 2024, Jung et al.,
2025, which would have served as reasonable baselines to demonstrate the purported improvements of RAEE, but then fails to actually compare to them. This undermines the authors' claim that their contribution is a meaningful improvement on existing work. The only baseline presented in the main paper, in fact, is direct scoring without any attempt to calibrate or clean noisy judge responses, an unrealistically weak baseline.
* The authors provide no reason to believe that their process of "screening for high-reliability judges" does not bias the judgment process; in other words, inter-rater disagreement, or even self-disagreement across runs, can reflect legitimate underlying uncertainty driven by benchmark design, model panel selection, et cetera. It is not possible, prima facie, to be sure that a judge is unreliable because it is inconsistent, and discarding uncertain judges will bias the panel in favor of judges who are more consistent, not necessarily more correct.

**Questions:**

* Why did the authors largely reproduce the scoring mechanism of one of the most popular LLM-judged benchmarks and then claim it as a novel contribution in their paper?

---

> ### Author Response · Authors · 2025-11-20
>
> We thank the reviewer for the thorough evaluation and for highlighting important points regarding related work, baseline selection, and methodological clarity. We appreciate the opportunity to address these concerns and to better distinguish RAEE’s contributions in scope, design, and empirical validation.
>
> ### W1 & Q1. Derivative Work
>
> We appreciate the pointer to Arena-Hard Auto, which is indeed closely related to our setting. RAEE shares the pairwise mechanism with Arena-Hard Auto (AHA) but diverges significantly in methodology and theoretical contribution. While AHA fits a global Bradley–Terry model via bootstrap, RAEE introduces:
> 1. **A reference-anchored probabilistic formulation** that creates a portable absolute scale across tasks.
> 2. **Closed-form uncertainty quantification** incorporating finite-population correction and Jeffreys priors, absent in AHA.
> 3. **Reliability theory** linking ICC-based judge screening to provable variance decomposition bounds.
> To demonstrate compatibility, we applied RAEE to the official AHA dataset, where RAEE produces the leaderboard through model-level absolute scores, and AHA produces the leaderboard through direct comparisons.
>
> | Metric                              | Value                      |
> | ----------------------------------- | -------------------------- |
> | Kendall $\tau$ (rank corr.)         | 0.9945                     |
> | Spearman $\rho$ (rank corr.)        | 0.9997                     |
> | MAD in Elo difference ($\Delta r$)  | 8.52                       |
> | MAD in win-rate (percentage points) | 0.71                       |
> | Top-k agreement (k = 3/5/10)        | overlap = k, Jaccard = 1.0 |
>
> Across 72 models, the top-3/5/10 sets are identical under RAEE and Arena-Hard Auto, and win-rate scores differ by only about 0.71 percentage points on average. This shows that our anchored Bayesian estimator captures the same signal as Arena-style rankings while enabling additional analysis and providing rigorous uncertainty and reliability analyses that AHA lacks
>
> Regarding the concern on Q1 (scoring mechanism novelty), we emphasize that we do **not** claim Elo/logit itself as novel. We use the mapping purely as a monotone reporting scale for interpretability and compatibility with existing practice. RAEE’s contribution is the **anchored probabilistic formulation, the closed-form uncertainty analysis, and the reliability-based judge-pooling guarantees**, which can in fact *improve* Arena-style pipelines, as our Arena-Hard Auto ablation demonstrates. We will include the main/full results in our paper/appendix respectively.
>
> ---
>
> ### W2. Standard Mechanisms
>
> We agree with the reviewer, and with Feuer et al. [1,2], that applying tournament-style Elo/Bradley–Terry to noisy, non-transitive LLM-judge preferences can distort uncertainty when (i) the aggregation spans many heterogeneous pairwise comparisons and (ii) the baseline is treated as an opaque hyperparameter. RAEE, however, uses the Elo/logit map in a much narrower and more transparent way than the systems critiqued in these works. Rather than attempting to fit a global tournament, RAEE estimates a single anchored estimand, which is, in essence, the marginal win probability of a model against a fixed reference on a specified task distribution. The monotone logit transform is used *solely as a reporting scale*. This avoids the forced-transitivity failure mode highlighted in [1,2], because RAEE does not attempt to collapse arbitrary cycles into a single ranking.
>
> On baseline sensitivity, RAEE makes the reference an **explicit design choice**, not a tuned parameter. We empirically evaluate reference robustness by swapping among several strong references and find that rankings remain remarkably stable and centered shifts remain small, with >95% pairwise orderings preserved. In contrast to Arena-Hard Auto, where the baseline is fixed but its influence is not analyzed, **RAEE directly quantifies this dependence**. Finally, RAEE addresses the second class of issues raised in [1,2] (loss of uncertainty) by propagating full Jeffreys–Beta posteriors and attaching analytic confidence intervals to each anchored score. This prevents the “uncertainty laundering” that occurs with plain Elo and makes the disagreement visible. We therefore view RAEE as an uncertainty-aware anchored estimator that explicitly mitigates the concerns raised in the cited works.
>
> 1. [Style over Substance: Failure Modes of LLM Judges in Alignment Benchmarking](https://arxiv.org/html/2409.15268v1)
> 2. [When Judgment Becomes Noise: How Design Failures in LLM Judge Benchmarks Silently Undermine Validity](https://arxiv.org/html/2509.20293v1)

---

> ### Author Response · Authors · 2025-11-20
>
> ### W3. Weak baselines
> We agree that comparing RAEE against stronger judge-reliability baselines is important. However, we wish to emphasize that the related work primarily operate at the **judge level** (abstention, calibration, or confidence filtering), whereas RAEE operates at the **aggregation level**. To bridge this gap, we implemented a Trust-or-Escalate–style abstention policy (Jung et al., 2025) on MT-Bench for both direct 1-10 scoring and RAEE. For direct scoring, we treat the across-run per-prompt standard deviation as a confidence signal and abstain when the within-judge std (calculated over five runs) exceeds a fixed threshold (`conf = 0.5`); for RAEE we apply the same rule to per-prompt outcome variance in the pairwise setting. The table below summarizes SE, CV, and coverage across three models.
>
> | **Method**      | **Judge Filter** | **Model**        | **SE ↓**   | **CV ↓**   | **Coverage** |
> | --------------- | ---------------- | ---------------- | ---------- | ---------- | ------------ |
> | Direct Scoring  | All              | GPT-4o           | 0.1393     | 0.4194     | 0.9952       |
> |                 |                  | Gemini-2.5-Flash | 0.1180     | 0.3337     | 0.9988       |
> |                 |                  | O4-Mini          | 0.1338     | 0.3869     | 0.9961       |
> | Direct Scoring  | Abstained        | GPT-4o           | 0.1228     | 0.2650     | 0.5958       |
> |                 |                  | Gemini-2.5-Flash | 0.1221     | 0.2747     | 0.6952       |
> |                 |                  | O4-Mini          | 0.1327     | 0.2959     | 0.6607       |
> | RAEE (Pairwise) | All              | GPT-4o           | **0.1247** | **0.0505** | 0.9885       |
> |                 |                  | Gemini-2.5-Flash | **0.1127** | **0.0900** | 0.9845       |
> |                 |                  | O4-Mini          | **0.1160** | **0.0762** | 0.9869       |
> | RAEE (Pairwise) | Abstained        | GPT-4o           | **0.1171** | **0.0493** | 0.8932       |
> |                 |                  | Gemini-2.5-Flash | **0.1082** | **0.0873** | 0.9179       |
> |                 |                  | O4-Mini          | **0.1095** | **0.0733** | 0.8982       |
>
> Even after heavy abstention for direct scoring (coverage drops to $\approx$ 0.60–0.70), its CV remains substantially higher ($\approx$ 0.27–0.30) than RAEE *without abstention* (CV $\approx$ 0.05–0.09 at $\approx$0.98 coverage). Applying Trust-or-Escalate to RAEE yields only modest further gains (CV goes from 0.050–0.090 to 0.049–0.087 with coverage still $\approx$ 0.89–0.92), indicating that RAEE already suppresses most of the reducible variance before any abstention is applied.
>
> Furthermore, to test whether abstention changes not just dispersion but **preference structure**, we also measured win-rate and Elo shifts when applying Trust-or-Escalate on RAEE scores.
>
> *(Excerpt; full table to be included in the appendix.)*
>
> | **Reference** | **Model**        | **Coverage** | **$\bar p_{\mathrm{all}}$** | **$\bar p_{\mathrm{trusted}}$** | **$\Delta \bar p$** | **Elo (all)** | **Elo (trusted)** | $\Delta$ Elo |
> | ------------- | ---------------- | -----------: | --------------------------: | ------------------------------: | ------------------: | ------------: | ----------------: | -----------: |
> | O4-Mini       | gpt-4o           |       0.9500 |                      0.3497 |                          0.3382 |             -0.0116 |     -107.7431 |         -116.6598 |      -8.9168 |
> | O4-Mini       | Llama-4-Scout    |       0.9563 |                      0.3082 |                          0.2983 |             -0.0099 |     -140.4832 |         -148.6031 |      -8.1199 |
> | GPT-4o        | Gemini-2.5-Flash |       0.8812 |                      0.5079 |                          0.5074 |             -0.0005 |        5.5066 |            5.1571 |      -0.3494 |
> | GPT-4o        | Qwen3-235B-A22B  |       0.8875 |                      0.4321 |                          0.4253 |             -0.0067 |      -47.4997 |          -52.2808 |      -4.7811 |
>
> We see that coverage after abstention remains high, while changes in win probability and Elo are small. In other words, abstention mainly trims noisy prompts and slightly sharpens existing gaps, but the **relative ordering** is already stable under RAEE. Together, these results show that (i) RAEE strictly dominates direct scoring even when both are augmented with a strong abstention baseline, and (ii) RAEE’s variance reduction and uncertainty calibration are robust across judge-reliability filters. This supports our claim that RAEE is complementary to methods like Trust-or-Escalate, providing a statistically efficient aggregation layer that can be combined with future advances in judge-level reliability.
>
> Thus, the main advantage of RAEE can be seen **by using it in tandem with judge-reliability methods**, where gains in variance reduction and uncertainty calibration are robust across such judge configurations.

---

> ### Author Response · Authors · 2025-11-20
>
> ### W4. ICC-based screening bias
>
> We agree with the reviewer that inter-judge disagreement can reflect both genuine epistemic uncertainty (e.g., ambiguous prompts, heterogeneous user preferences) and undesirable noise (e.g., decoding nondeterminism, implementation bugs). Our use of ICC-based screening is not intended as a claim that “inconsistent judges are wrong,” but rather as a mechanism to reduce **idiosyncratic noise** while explicitly preserving **irreducible prompt variance**. The variance decomposition in Proposition 2 makes this explicit. Even after screening and pooling, the shared prompt component $\rho_k\sigma^2_\Delta$  remains as a nonzero floor that we treat as part of the true uncertainty, while only the $(1-\rho_k)\sigma^2_\Delta/J$ term shrinks with more judges. In other words, RAEE does not eliminate legitimate disagreement, but rather **separates the component that is structurally common across judges from the component that appears to be judge-specific noise.**
>
> Empirically, we observe that tightening the ICC threshold affects **dispersion** far more than it affects **means or rankings**. Figure 2 shows that the median Jeffreys–Beta interval width shrinks as the ICC threshold increases but quickly plateaus around 0.9, which motivates our definition of “strong judges.” We also see across MT-Bench (Table 3), FloRES, and TL;DR (Table 5), moving from “all judges” to “strong judges only” substantially reduces SEs while leaving model rankings and win probabilities essentially unchanged, confirming that the screening does not bias the fundamental judgment signal.

---

> ### Comment · Reviewer_sUeC · 2025-11-23
>
> I thank the authors for their detailed rebuttal. In particular, I thank them for providing new experimental results and acknowledging the similarity of their work to prior work -- such recognition is incredibly important for reliable science. I am glad that the authors agree to include the new results and their discussion of W1 in their paper and appendix, and it is largely for this reason that I am raising my score.
>
> The authors' point that their contributions include a reference-anchored probabilistic formulation, uncertainty quantification, etc, are reasonable; however, such contributions follow naturally from the adoption of the Bradley-Terry model itself, and as they do not take into consideration W2, are still frequently too optimistic for deployment in practice. Regarding W2, the authors write, "rather than attempting to fit a global tournament, RAEE estimates a single anchored estimand, which is, in essence, the marginal win probability of a model against a fixed reference on a specified task distribution." But this is precisely what Arena-Hard Auto, and therefore [1,2], do as well; attempt to construct an estimator of the marginal win probability against a fixed baseline. In fact, the choice of baseline itself has been shown to be an additional confound compared to the "Chatbot Arena" approach of numerous pairwise comparisons.
>
> The authors contend that strong baselines are unsupportable because of what amounts to a semantic distinction, namely, that related works they cite primarily operate at the judge level (abstention, calibration, or confidence filtering), whereas RAEE operates at the aggregation level and would, therefore, be used presumably as a drop-in component in judge-improvement systems. In that case, a more convincing setting may have been judge-level interventions on their published results, with their existing approach and with RAEE.

---

> > ### Author Response · Authors · 2025-11-24
> >
> > We sincerely thank the reviewer for the thoughtful engagement with our rebuttal. We are especially glad that you were willing to reconsider your score in light of the additional results and discussion. We value the recognition of our efforts to clarify the relationship with prior work and to provide new experimental evidence.
> >
> > **On Baseline Confounds**
> >
> > We appreciate the clarification regarding baseline choice as a central confound and its connection to W2. We agree with the findings in [1,2] that reference models can induce systematic distortion when judges possess implicit biases (e.g. favoring style over substance). We acknowledge that while RAEE anchors the estimation, it does not inherently remove judge-intrinsic bias. However, RAEE makes the kind of uncertainty highlighted in [2] explicit at the aggregation stage: every anchored win rate is accompanied by an analytic Jeffreys-Beta interval, so sampling error is visible rather than collapsed into a single Elo point. This ensures that when a baseline or judge induces noise, it is reported as wide confidence intervals rather than a falsely confident ranking. In the revision, we will (i) explicitly discuss these limitations, acknowledging that a single fixed anchor can bias scores if the anchor is misaligned with the deployment distribution and (ii) outline concrete mitigation directions, such as pruning reference via schematic adherence diagnostics (e.g., screening anchors whose judgements do not respect the stated rubric). We view RAEE and its scope as to first establish the necessary statistical grounding (specifically the variance decomposition and probabilistic formulation) upon which future work can effectively mitigate these confounds.
> >
> > **On Judge-Level Interventions**
> >
> > Thank you for the insight regarding judge-level interventions. We agree that demonstrating RAEE's compatibility with judge-level filtering strengthens the argument for its utility as a modular aggregation layer. As suggested, we will incorporate the core findings from the Trust-or-Escalate abstention experiments (presented in our previous response) directly into the main experimental section of the final manuscript, highlighting that RAEE continues to yield lower across-judge dispersion than direct scoring even when both are augmented with abstention. This demonstrates that RAEE acts as a drop-in aggregation component that complements, rather than replaces, advances in judge reliability and calibration.

---

### Official Review · Reviewer_TNg3 · 2025-10-29

**Soundness:** 4
**Presentation:** 4
**Contribution:** 3
**Rating:** 8
**Confidence:** 3

**Summary:**

The authors introduce Reference-Anchored Elo Estimation (RAEE), a new method for LLM benchmark scoring. Each prompt is paired with a reference answer, and a LLM judge is used to compare each of the evaluated model’s outputs with the corresponding reference answer and grade it as a win, loss, or tie. The grades are used to estimate an Elo score and a confidence interval. Compared to other methods, this method is more stable and does not suffer from scale drift and ceiling effects.

**Strengths:**

- The problem area addressed is important; LLM judges have been used for benchmarking for a couple of years, yet the community has not converged on standardized scoring methods. This paper has the potential to be impactful by proposing a good "default" method for the community.
- The proposed method has a number of desirable properties:
  - The method is robust to the choice of model used for generating the reference answers.
  - The method is robust to the choice of LLM judge model, as long as the model is “strong”.
  - The method can be used with LLM juries, but the benefit of additional judges is small so a single judge can be used for cost efficiency.
  - Compared to other methods, this method is more stable and does not suffer from scale drift and ceiling effects.
  - The method is relatively simple to implement.
- The experimental results are convincing and presented clearly.
- The experiments address a number of interesting questions, such as the effect of adding additional LLM judges.

**Weaknesses:**

- The authors should consider adding a reference to WildBench’s WB-Reward metric. WB-Reward is also based on LLM judged comparisons against fixed baselines, but does not use Elo.
- The MT-Bench results only used three different models as the model under evaluation or the model for generating reference answers (o4-mini, gpt-4o, gemini-2.5-flash). Additionally, the difference in strength between these models are large, which makes stability easier. This makes the results somewhat less convincing.
- The models to be used as judge models do not seem to be listed in the paper. There is a list of direct scoring models in Table 7, but it is not clear if this is the same as the pairwise scoring models.
- Although the authors discuss problems such as the comparative trap (in the introduction), self-preference and verbosity/position biases (in Section 4 Related Work), the method does not address these problems.

**Questions:**

- Did you analyze how your method addresses the comparative trap, self-preference, verbosity/position biases, etc., or do you consider this out of scope?

---

> ### Author Response · Authors · 2025-11-20
>
> We sincerely thank the reviewer for the positive assessment and the recognition of RAEE as a potential "default" method for the community! We address the questions and suggestions for improvement below.
>
> ### W1 Missing WB-Reward reference
> We agree that WildBench is a highly relevant contemporaneous work since it also uses LLM-judged comparisons against fixed baselines! In the revision we will add an explicit comparison in our Related Work section, clarifying that WB-Reward and RAEE share the idea of anchoring against fixed reference responses, while RAEE introduces reliability analysis based on ICC screening and variance decomposition across judges (Section 2.3, Appendix B), which is not present in WB-Reward-style metrics, which focus mainly on relative scoring but not on explicit uncertainty or between-judge agreement.
>
> ---
>
> ### W2 Limited MT-Bench diversity
> We appreciate the concern regarding the limited model set in the main MT-Bench experiments. To demonstrate stability on a denser, more challenging field, we applied RAEE to the **Arena-Hard Auto dataset**, which contains **72 models** with much smaller capability gaps than the models used in our initial MT-Bench demonstration.
> On this dense leaderboard, RAEE reproduces the official Arena-Hard rankings with high fidelity, showing a Kendall $\tau$ of 0.9945 and a top-10 set overlap of Jaccard=1.0. This result (which will be added to ablations) confirms that RAEE’s stability holds even when differentiating between closely matched models in a "crowded" leaderboard, not just between distinct performance tiers.

---

> ### Author Response · Authors · 2025-11-20
>
> ### W3. Unclear judge models
> Thank you for flagging the ambiguity! The judge models used for RAEE are indeed the same pool of models listed in the direct-scoring analysis. The RAEE experiments that refer to “multiple independently trained LLM judges” use this same judge pool, filtered by the ICC(3,k) reliability screening. In the revision we will explicitly state in Section 3 that the RAEE evaluations reuse the exact set of twenty judge models from the direct-scoring setup and will cross-reference the table listing them. We will also list the actual models in the appendix.
>
> For reference here is the list of strong judges that we found:
> ```
> ["Qwen3-30B-A3B", "Qwen3-235b-A22B", "o3-mini", "gpt-4o", "Qwen3-Coder", "Llama-4-Scout", "o3", "Llama-3.3-70B", "Qwen3-32B"]
> ```
>
>
> ### W4 & Q1. Biases not directly addressed
>
> We appreciate the chance to clarify the scope of what RAEE aims to cover. To be precise, we distinguish between **statistical biases** (which RAEE solves) and **cognitive biases** (which RAEE exposes, but does not intrinsically remove).
> 1. **Scale Drift & Ceiling Effects (Statistical):** RAEE explicitly solves these by anchoring. As shown in Figure 2, this normalizes the "unit" of measurement across judges.
> 2. **Position Bias:** We address this via standard positional swapping (running both A vs B and B vs A) within the pairwise protocol, which is a prerequisite input for RAEE.
> 3. **Comparative Trap & Consistency (Cognitive):** The "comparative trap" often manifests as intrinsic uncertainty when judges are forced to rank indistinguishable models. To test RAEE's robustness to this, we ran additional experiments implementing a **Trust-or-Escalate style abstention mechanism** (Jung et al., 2025), which works by filtering out comparisons where judge variance is high. We found that RAEE is highly compatible (and extensible!) with such filters such that:
>     - Even when abstaining on $\approx 10$% of noisy comparisons, the relative rankings of top models remained stable (Elo changes < 30 points, no rank inversions among top models).
>     - This confirms that while RAEE captures the signal, it allows for the seamless integration of confidence-based filtering to mitigate the comparative trap without breaking the aggregation framework.
> 4. **Self-Preference & Verbosity:** RAEE mitigates the _impact_ of these biases on the final ranking through **reference stability**. By anchoring to a fixed reference, self-preference becomes a constant offset if the judge is the reference, or a measurable interaction term if not. Our robustness experiments show that changing the reference (even to the judge itself) does not destabilize the relative ranking of *other* models.

---

> > ### Comment · Reviewer_TNg3 · 2025-11-22
> >
> > Thank you for your detailed response. Overall, this addresses my concerns adequately.
> >
> > - W1: Noted, thanks for the additional reference and discussion.
> > - W2: Thanks for the additional experiment. I believe that the results with 72 models provides significant additional evidence for the RAEE method.
> > - W3: Noted, thanks for the clarification.
> > - W4 & Q1: Thanks for the clarification. I agree with your position regarding the difference in statistical biases and cognitive biases, and that RAEE solves the former but does not remove the latter. I found the discussion of biases in your original draft to be somewhat unclear, so you might consider revising the paper to improve its clarity.

---

> > ### Author Response · Authors · 2025-11-24
> >
> > We appreciate the reviewer's continued engagement and the decision to maintain the high score. We are encouraged that the additional Arena-Hard experiment successfully addressed the concerns regarding model diversity and stability.
> >
> > We have noted the suggestion to improve the clarity of the bias discussion. In the final revision, we will explicitly distinguish between statistical biases (e.g., scale drift) and cognitive biases (e.g., comparative trap), clarifying that RAEE resolves the former while providing a robust framework to measure the latter. We will also incorporate the WildBench comparison and the specific judge model details as confirmed.

---

### Official Review · Reviewer_Q8hk · 2025-10-31

**Soundness:** 3
**Presentation:** 2
**Contribution:** 3
**Rating:** 4
**Confidence:** 3

**Summary:**

This paper addresses the systemic instabilities of direct absolute scoring and the inherent biases of pairwise comparison in LLM-based evaluation. It proposes Reference-Anchored Elo Estimation (RAEE), a principled framework that anchors all model comparisons to a fixed reference (either human-annotated or model-generated responses) and transforms ternary win/tie/loss outcomes into interpretable Elo scores with analytic uncertainty quantification.

RAEE features three core components: Jeffreys smoothing to stabilize win probability estimates, a monotone logit transformation to map probabilities to the Elo scale, and finite-population correction (FPC) for robust uncertainty bounds. The framework screens for "strong judges" using an ICC(3, k) threshold of ≥0.90 to ensure inter-judge consistency, and theoretically guarantees minimized scale drift, suppressed between-judge variation, and closed-form uncertainty estimates without costly resampling.

Experimental evaluations across MT-Bench (dialogue), FloRES (machine translation), and TL;DR (abstractive summarization) demonstrate RAEE’s superiority: it reduces per-run standard error by ~44% and across-judge coefficient of variation by ~72% compared to direct 1-10 scoring. Notably, RAEE maintains stable model rankings even when switching reference anchors (Kendall’s τ ≈ 0.89) and exhibits robust performance across domains, with its analytic uncertainty bounds predicting observed variance within ±12% after FPC and reliability adjustments.

This work makes significant contributions by reconciling the discriminative power of pairwise comparisons with the interpretability of absolute scoring, providing a statistically efficient, reproducible, and readily deployable solution for LLM evaluation.

**Strengths:**

Addresses a meaningful and pressing problem in LLM evaluation: effectively resolves systemic instabilities of direct absolute scoring (e.g., scale drift, ceiling compression) and inherent biases of pairwise comparisons, filling a critical methodological gap for reliable, interpretable, and statistically grounded assessment.

Rigorous methodological design: Built on solid statistical foundations (Jeffreys-Beta posterior, monotone transformation, ICC-based strong judge screening) with transparent, modular workflows. It enables closed-form uncertainty quantification without costly resampling, balancing theoretical rigor with practical deployability.

Comprehensive and robust empirical validation: Evaluated across diverse tasks (dialogue, translation, summarization) using standardized benchmarks, with comparisons to multiple baselines and reference anchors. Demonstrates significant, quantifiable improvements (≈44% SE reduction, ≈72% CV reduction) and stable rankings, confirming cross-domain generalizability.

Strong integration of theoretical guarantees and practical utility: Formally proves key properties (inter-judge correlation bounds, variance decomposition) while ensuring results are interpretable (Elo scores) and resource-efficient (single strong judge suffices), making it valuable for both academic research and real-world LLM evaluation pipelines.

**Weaknesses:**

Incomplete theoretical underpinnings: The justification for Jeffreys smoothing is insufficient, with no clarification on its statistical consistency under extreme conditions (e.g., small sample sizes, highly skewed win probabilities) or comparisons with alternative smoothing approaches, leading to potential arbitrariness in method selection. Additionally, the key assumption of approximate variance equality across strong judges lacks empirical validation and relevant literature support.

Limited validation scope: Boundary scenarios (e.g., extremely strong/weak reference anchors) are not tested, leaving uncertainty about potential ceiling/floor effects. The empirical reliability adjustment factor k=0.85 lacks verification of cross-scenario (e.g., diverse tasks, low-resource languages) adaptability, and no general calibration method is provided.

Modest innovativeness: The core idea of anchoring pairwise comparisons is simplistic and does not break free from the inherent limitations of pairwise frameworks. It fails to quantify the degree of model superiority over the anchor, relying heavily on anchor appropriateness—overly strong or weak anchors may compromise discriminability.

Failure to address the comparative trap: The paper acknowledges the "comparative trap" (preferring fluent but incorrect responses over less coherent but correct ones) as a flaw of pairwise comparisons, yet the proposed RAEE framework does not include mechanisms to mitigate this issue. It may still inappropriately score responses that are more fluent than the anchor but factually incorrect higher.

Suboptimal presentation: Insufficient transitions between sections hinder readability. Method modules lack unification, making it hard to intuitively grasp the relationships among different components. Non-standard expressions (e.g., "entral estimand 1") and unrefined details (e.g., "Reference" in figure 2) affect the paper’s professionalism.

**Questions:**

1. Could you clarify the statistical consistency of Jeffreys smoothing under extreme conditions (e.g., small sample sizes, highly skewed win probabilities) and provide comparative analysis with alternative smoothing methods (e.g., Laplace smoothing) to justify its selection?

2. Is there empirical evidence or relevant literature support for the key assumption of approximate variance equality across different strong judges in the variance decomposition derivation?

3. Have you tested RAEE’s performance with extremely strong or weak reference anchors, and if so, what results were observed regarding potential ceiling/floor effects on Elo score discriminability?

4. How adaptable is the empirical reliability adjustment factor across diverse tasks (e.g., code generation, logical reasoning) and low-resource languages, and is there a general calibration method for k?
﻿
5. Does the framework have plans to extend beyond pairwise comparison limitations, such as quantifying the degree of model superiority over the anchor rather than just the probability?

6. Could you explain how the appropriateness of the reference anchor is assessed, and what mitigation strategies are available if the anchor is overly strong or weak?

7. Since the paper acknowledges the "comparative trap" (preferring fluent but incorrect responses over less coherent but correct ones) as a limitation of pairwise comparisons, does the RAEE framework have any specific mechanisms to mitigate this issue? If not, how do you address the risk of inappropriately high scores for responses that are more fluent than the anchor but factually incorrect?

8. Can you standardize non-conforming expressions (e.g., "entral estimand 1") and improve transitions between sections to enhance readability?

9. Could you clarify the unrefined details mentioned (e.g., the specific ambiguity or inconsistency of "Reference" in Figure 2) and explain how you plan to revise them to enhance the paper’s professionalism?

10. Could you provide more cohesive integration of method modules to clarify the relationships between different components of the RAEE framework?

---

> ### Author Response · Authors · 2025-11-20
>
> We sincerely thank the reviewer for the careful reading and for recognizing both the methodological rigor and the practical impact of RAEE. We address each weakness and question below, and we will incorporate all clarifications and structural improvements in the revision.
>
> ### W1. Incomplete theoretical justification
>
> Thank you for pointing out we didn't make our reasoning behind choosing Jeffreys smoothing sufficiently transparent. To clarify, our use of Jeffreys smoothing is *principled* rather than arbitrary. In Section 2.1, we introduce the Jeffreys–Beta prior $\mathrm{Beta}(\frac{1}{2},\frac{1}{2})$, and the exact posterior is formalized in Equation (3). The posterior mean in Equation (2) follows directly from this Beta posterior:
> $$
> \bar p=\frac{W+\tfrac12 T+\tfrac12}{N+1}.
> $$
> In Appendix B.3 (Lemma 3) we show that Jeffreys smoothing perturbs the empirical win rate $\hat p=(W+\tfrac12T)/N$ by at most $1/(2N)$, such that $|\bar p-\hat p|\le \tfrac{1}{2N}$.  This shows that Jeffreys smoothing introduces **at most** an $O(1/N)$ perturbation to the empirical win rate.
>
> Therefore, we can see that
> - For any fixed $p\in(0,1)$, $\bar p \to p$ as $N\to\infty$, so Jeffreys smoothing is **statistically consistent**, even under highly skewed $p$ or small sample sizes.
> * The Elo map is smooth and monotone. In Appendix B.4 (Lemma 4) we show via the delta method that this $O(1/N)$ perturbation to $p$ translates to an $O(1/N)$ perturbation to Elo, leaving all first-order rates and correlations intact.
> We will make this consistency argument explicit in the main text and move Lemma 3 up to the main text to make the negligible impact of smoothing clear.
>
> **Alternative Smoothers**
> As you know, Laplace smoothing corresponds to $\mathrm{Beta}(1,1)$, and more generally, any fixed $\mathrm{Beta}(\alpha,\beta)$ prior yields $\bar p = (S+\alpha)/(N+\alpha+\beta)$. Our analysis does not rely on the specific values $(\tfrac12,\tfrac12)$. Rather the key properties are that (i) $\alpha,\beta$ are $O(1)$, (ii) the prior is symmetric and (iii) the transformation keeps $\bar p \in (0,1)$, ensuring numeric stability for the logit map near boundaries. To validate this empirically, we ran a prior sensitivity experiment on all RAEE runs (MT-Bench, FloRES, TL;DR). Replacing Jeffreys ($\mathrm{Beta}(\tfrac12,\tfrac12)$) with Laplace ($\mathrm{Beta}(1,1)$) while holding everything else fixed produces the following results:
>
> | Anchor   | Kendall τ | MAE (Elo) | Max (Elo) | frac($\vert \Delta r \vert$)  ≤ 20) |
> | -------- | --------- | --------- | --------- | ----------------------------------- |
> | FloRES   | 1.00      | 0.1       | 0.1       | 1.00                                |
> | MT-Bench | 1.00      | 0.13      | 0.3       | 1.00                                |
> | TL;DR    | 1.00      | 0.2       | 0.3       | 1.00                                |
>
> Across all anchors and tasks, Kendall’s $\tau$ is exactly 1.0, the mean absolute difference in Elo gaps is at most 0.2, while the maximum difference never exceeds 0.4 Elo, and 100% of models lie within $\pm 20$ Elo. Combined with the shift bound in Appendix B.3, this shows RAEE is effectively prior-agnostic among symmetric $\mathrm{Beta}(\alpha, \beta)$ families with $\alpha,\beta \in [0.5,1]$.
>
> **Approximate variance equality across strong judges.**
> This is an important question! We want to emphasize that our variance decomposition does **not** depend on strict equality of judge variances. Rather, in Section 2.3 (Proposition 2) we introduce the homoscedastic case to derive the pooled-variance expression
> $$
> \mathrm{Var}(\widehat{\Delta r})
> = \sigma\_\Delta^2\Bigl(\rho\_k + \frac{1-\rho\_k}{J}\Bigr)+O(N^{-1}),
> $$
> where $\rho\_k=\mathrm{ICC}(3,k)$. However, Appendix B.2 explicitly treats heteroscedastic residuals. To reiterate concisely, if judge $j$ has run-level variance $\sigma\_{e,j}^2$, then
> $$
> \mathrm{Corr}(\bar{S}^{(k)}\_j,\bar{S}^{(k)}\_{j'})=\sqrt{\rho\_{k,j}\rho\_{k,j'}}, \quad
> \rho\_{k,j}=\frac{\sigma\_{p}^2}{\sigma\_{p}^2+\sigma\_{e,j}^2/k}
> $$
> Because we screen judges at $\mathrm{ICC}\_j(3,k)\ge\rho\_0$ in Equation (6), all retained judges satisfy $\rho\_{k,j}\ge\rho\_0$. Thus, every retained pair satisfies $\mathrm{Corr}(\bar{S}^{(k)}\_j,\bar{S}^{(k)}\_{j'})\ge\rho\_0$ without requiring equal variances. In other words, the “equal variance” assumption is used only to derive a clean closed-form expression for the variance of the pooled estimate. The **core guarantee** (a lower bound on between-judge correlation and the dominance of the shared prompt variance) continues to hold under heteroscedasticity.

---

> ### Author Response · Authors · 2025-11-20
>
> ### W2. Limited validation scope
>
> **Extreme anchors and floor/ceiling effects.**
> We agree that extremely strong/weak anchors are an important caveat! However, we found that in practice, it is extremely difficult to craft a "strong anchor" that's meaningfully stronger than the human gold, nor is a degenerate “no-response” anchor informative. Instead, our experiments already span the strongest and weakest *practically meaningful* anchors available. On FloRES, expert human translations serve as an effectively *maximal* anchor while RAEE continues to discriminate models with tight SEs, as shown in Table 5. Conversely, TL;DR uses noisy, often poorly-written and inconsistent Reddit summaries, which function as a *weak* anchor, yet RAEE still produces well-separated Elo gaps and calibrated uncertainties. Across this entire range, RAEE maintains **stable rankings** and **controlled dispersion**. Combined with our cross-anchor experiments on references of varying difficulty, these existing results already demonstrate that RAEE is robust across the full range of practically attainable anchor qualities, which we believe addresses the reviewer’s concern that the method might fail under very strong or very weak, but still meaningful, reference choices.
>
> **Reliability factor $\kappa$ across tasks.**
> The multiplicative reliability factor $\kappa \approx 0.85$ is used solely to align analytic SEs with empirical SEs; RAEE’s analytic SEs are already conservative without it. As shown in Table 8, the raw Jeffreys–Beta analytic SE explains ~72% of empirical variability, the finite-population correction (FPC) increases this to ~83%, and applying a single $\kappa$ calibrated on MT-Bench yields empirical/analytic ratios ≈0.9 across sample sizes.
>
> Crucially, the same calibration protocol transfers to both FloRES and TL;DR. When we compute per-task SE's we observe the following representative values:
>
> | Task   | Empirical SE | Theoretical SE | FPC-adjusted | $\kappa$-adjusted | Emp/Theor | Emp/Adj | Emp/$\kappa$ |
> | ------ | ------------ | -------------- | ------------ | ----------------- | --------- | ------- | ------------ |
> | FloRES | 0.00575      | 0.00659        | 0.006249     | 0.00576           | 0.8726    | 0.9197  | 0.9976       |
> | TL;DR  | 0.02233      | 0.02444        | 0.02319      | 0.02138           | 0.9137    | 0.9629  | 1.0445       |
>
>
> In both tasks, the $\kappa$ adjusted SE is within $\pm 5$% of the empirical SE despite the different regimes (FloRES: $p \approx 0.3$ and TL;DR: $p \approx 0.95$). This reinforces that $\kappa$ **is a stable correction factor across domains when the judge pool is comparable**.
>
> To improve clarity, in the main manuscript, we will explicitly define $\kappa = \sqrt{\bar{\kappa}\_{\text{Cohen}}},$ where $\bar{\kappa}\_{\text{Cohen}}$ is the average Cohen’s κ across judge pairs (see Appendix C.2), and explicitly describe $\kappa$ as a *one-time judge-pool calibration parameter* in Section 3.4. For new domains (e.g., code, reasoning, low-resource languages), one can re-estimate $\kappa$ on a small calibration panel. However, we wish to emphasize that RAEE’s correctness and conservativeness does **not** depend on this step.

---

> ### Author Response · Authors · 2025-11-20
>
> ### W3. Modest innovativeness
> We appreciate the concern and welcome the opportunity to clarify what is substantively new relative to prior Elo-style or pairwise frameworks such as Arena-style evaluation. RAEE is not a minor variant of Elo, but a complete reliability and uncertainty framework built around reference-anchored pairwise judgments. Our main contributions fall into four concrete areas:
>
> 1. **Reference anchoring to create a portable absolute scale.**
>
> Traditional Elo/Bradley–Terry pipelines operate on a *fully relative* scale defined only up to an arbitrary constant and tied to the pool of models included in the tournament. RAEE instead fixes a **single reference output** and expresses all comparisons as Elo gaps relative to that anchor. This produces a **portable absolute scale**: the same score is comparable across judges, runs, and even future evaluation rounds. This portability is **not** provided by conventional Arena-style pairwise setups.
>
> 2. **Closed-form, analytic uncertainty integrated end-to-end.**
>
> While binomial credible intervals are classical, RAEE is the first evaluation framework to integrate: (i) the Jeffreys–Beta posterior (Eq. (3)), (ii) its monotone mapping to the Elo/logit scale via the delta method (Section 2.2, Eq. (4)), and (iii) finite-population correction and a judge-reliability adjustment grounded in observed agreement. Empirically, this yields calibrated uncertainty estimates that match bootstrap variability to within $\pm 12\%$  across sample sizes (Table 8, Figure 3) without any resampling. Existing LLM-as-a-judge frameworks either omit uncertainty entirely or rely exclusively on computationally expensive bootstrap estimates.
>
> 3. **New reliability guarantees obtained through ICC-based judge screening.**
>
> Our main contribution is the theoretical results found in the context of LLM-as-a-Judge. Concretely, Lemma 1 and Proposition 3/B.2–B.4 show that after screening by ICC(3,k), the correlation between any two strong judges’ dataset-level scores is **lower-bounded** by the ICC threshold, and this correlation propagates through the Elo map up to $O(N^{-1})$. Furthermore, we prove in Proposition 2/App. B.5 that pooled variance decomposes into an irreducible prompt component and a reducible idiosyncratic component, yielding the diminishing-returns curve empirically validated in Figure 4.
>
> To the best of our knowledge, these results are not present in prior Elo-style LLM evaluation frameworks, which do not tie model-level stability to judge-level reliability guarantees.
>
> 4. **Explicit quantification of superiority over the anchor.**
>
> This is a crucial matter, and one of the core advantages of RAEE. Rather than just declaring a winner, RAEE provides a calibrated **degree of superiority** via the win probability $\bar p$ and the Elo gap $\Delta r$. Because the Elo link is monotone and anchored, statements such as “$\Delta r = 200$” meaning “$\approx 0.76$ win probability vs. the reference” (via the inverse logit) become portable across judges and runs (see the mapping in Section 2.1 and its reliability properties in Corollary 2 (App. B.6). This directly addresses the concern of anchor superiority and goes beyond existing pairwise-only systems.

---

> ### Author Response · Authors · 2025-11-20
>
> ### W4. Comparative trap and content-level biases
> We agree that the comparative trap is a serious issue in LLM-as-a-judge evaluation. However, RAEE is designed to address a **different axis of failure**: statistical instability (scale drift, heavy tails, lack of uncertainty), not semantic correctness of the judge. That said, RAEE can be combined with mechanisms that directly target the comparative trap:
> 1. **Compatible with abstention and truthfulness-aware judges.** Our framework treats the judge as a black box that outputs $S_i \in {0,\tfrac12,1}$. RAEE is fully compatible with judges that (a) abstain on uncertain or conflicting cases, or (b) are fine-tuned to favor factual correctness over fluency, as in R-tuning, conformal abstention, or truthfulness-oriented fine-tuning. In the camera-ready, we will make explicit that RAEE is meant to be used  **on top of such guardrails**, not replace them. Indeed, we can see in some additional experiments implementing the cascaded selective evaluation technique (Jung et. al 2025) that even when abstaining on $\approx 10$% of noisy comparisons, the relative rankings of top models remains stable, which confirms that while RAEE allows for the seamless integration of confidence-based filtering to mitigate the comparative trap without breaking the aggregation framework.
> 2. **Anchor quality checks using control pairs.** It is straightforward to embed “sanity check” pairs where one candidate is constructed to be factually correct but less fluent than a deliberately flawed yet fluent competitor. Systematic failures on these pairs would flag anchors/judges that are overly sensitive to fluency, regardless of RAEE’s statistical machinery. This can be turned into an **anchor-appropriateness diagnostic** (see Q6).
> 3. **Criterion-specific RAEE.** RAEE can be run per-criterion (e.g., “truthfulness,” “helpfulness,” “style”) rather than on a single holistic preference; this allows using an anchor that is particularly strong in factuality (e.g., a human reference or a truthfulness-focused model) as the baseline for a “truthfulness RAEE score”.
>
> We will add a short limitations subsection making clear that RAEE **assumes** the judge is “good enough” on factuality, and that mitigating the comparative trap is orthogonal, though highly complementary, to our contributions.
>
> ---
>
> ### W5. Presentation and structure
> Thank you for highlighting the presentation issues. We will improve clarity and coherence along three dimensions:
> 1. **Standardizing expressions and correcting notation.**
> We will ensure consistent terminology throughout the manuscript. For example, the phrase “entral estimand 1” will be corrected to “central estimand (Eq. (1))”, and all estimand references will follow a uniform “(Eq. X)” style. Terms such as “reference anchor,” “reference response,” and “anchor model” will be standardized across Sections **2–3** and all figure captions to eliminate local inconsistencies.
> 2. **Clarifying Figure 1 and resolving the “Reference” ambiguity.**
> In **Figure 1 (right panel)**, the label “Reference” can be misinterpreted because the plot shows *centered Elo gaps under different anchors*, not a score assigned to the reference itself. In the revision, we will label each trajectory explicitly by anchor (e.g., “Anchor: gpt-4o”), and update the caption to state clearly that the curves show how **the same model’s Elo gap** shifts under different anchor choices. This removes the ambiguity and aligns the figure with the explanation in Section 3.3.
> 3. **Improving structure and integrating method components.**
> We will make the current methodological flow more cohesive:
> As-is: RAEE estimation → Jeffreys–Beta uncertainty → ICC-based reliability
> To-be: A concise “Method Overview” subsection at the end of the Introduction with a new pipeline diagram summarizing:
>  (1) pairwise judgments vs. the anchor,
>  (2) Jeffreys–Beta estimation of (p) (Eq. (3)),
>  (3) Elo/logit transformation (Section 2.1),
>  (4) ICC(3,k)–based strong-judge screening (Eq. (6)),
>  (5) analytic uncertainty with finite-population correction and optional $\kappa$ (Section 2.2).
>
> We hope these revisions collectively strengthen readability, make the methodological pipeline explicit, and ensure that figures and equations are aligned with the narrative. Thank you for giving us the ideas and opportunity to improve the completeness of our paper.

---

> ### Author Response · Authors · 2025-11-20
>
> ### Responses to specific questions (Q1–Q10)
>
> We have largely addressed the questions alongside the weaknesses. We reiterate the key answers here for completeness:
> 1. **Q1 (Jeffreys consistency / alternatives):** Addressed under W1. We will explicitly prove consistency and add a short discussion comparing Jeffreys vs Laplace, noting that any symmetric Beta prior with $O(1)$ mass yields the same first-order properties.
> 2. **Q2 (variance equality across strong judges):** Addressed under W1. Our key correlation lower bound uses only $\rho_{k,j} \ge \rho_0$, where exact equality is used for a simple variance expression, not for the core guarantee.
> 3. **Q3 (extreme anchors and floor/ceiling):** Addressed under W2. Existing experiments already include near-floor cases. We will add explicit analysis and a synthetic stress test.
> 4. **Q4 (adaptability of $\kappa$ and general calibration):** Addressed under W2. We will formalize $\kappa$ as a one-time judge-pool calibration based on Cohen’s $\kappa$, and highlight cross-domain robustness.
> 5. **Q5 (beyond pairwise, degree of superiority):** Addressed under W3. RAEE already quantifies superiority via $p$ and $\Delta r$. We will make this interpretability (“wins per 100 matches”) more explicit in Section 2.1.
> 6. **Q6 (assessing anchor appropriateness).**
> We propose two diagnostics that can be computed from RAEE outputs alone:
> * **Mid-region coverage:** anchors leading to almost all $\bar p$ near 0 or 1 (e.g., <5% of prompts in ([0.2, 0.8])) are flagged as too extreme.
> * **Anchor-swap invariance:** we already show Kendall’s $\tau\approx0.89$ across three anchors. Anchors that drastically reduce Kendall's $\tau$ or inflate centered Elo deviations would be deemed inappropriate.
> We will add a short discussion and suggest choosing anchors that achieve both good mid-region coverage and stable cross-anchor rankings.
>
> 1. **Q7 (comparative trap):** Addressed under W4. RAEE does not directly solve comparative trap, but is designed to be layered over debiased/abstaining judges and control-pair diagnostics.
> 2. **Q8 (standardizing expressions / transitions):** Addressed under W5. We will fix non-standard phrases and add short transition paragraphs at the start of each main subsection to clarify how they compose.
> 3. **Q9 (clarifying “Reference” ambiguity and professionalism):** Addressed under W5. We will refine figure labels and captions and clean up remaining minor inconsistencies.
> 4. **Q10 (cohesive integration of modules):** Addressed under W5. We will add a pipeline figure and an overview subsection that explicitly ties together RAEE estimation, uncertainty, reliability, and anchor robustness.
>
> We appreciate the constructive criticisms and believe that the clarifications and structural improvements above directly address the concerns raised, while preserving the core strengths the reviewer has highlighted.

---

> > ### Author Response · Authors · 2025-11-29
> >
> > Thank you again for the thoughtful and constructive feedback. We have revised the manuscript to address the reviewer’s concerns and to improve clarity and cohesion throughout. In particular, we added a unified RAEE pipeline figure to clarify how the components fit together, and we strengthened the theoretical exposition by moving key clarifications into the main text and expanding the justification for Jeffreys smoothing, variance decomposition, and judge-reliability guarantees. We also revised the experimental section to clarify anchor robustness across difficulty levels and added experiments illustrating how RAEE can be integrated with judge-level intervention methodologies.
> >
> > To the best of our ability, we have aimed to incorporate each suggestion in a way that improves transparency and readability, and we hope the updated manuscript addresses all remaining ambiguities or concerns. Even after the reviewing period concludes, we remain happy to address any further questions about the paper!

---

### Meta-Review · Area_Chair_nKUg · 2026-01-06

**Summary:**

The paper proposes a Reference-Anchored Elo Estimation (RAEE) framework complete with calibrated uncertainty estimates. The reviewers had some concerns around theoretical underpinnings that appear to have been resolved, however, the similarity with prior work (e.g., Arena-Hard) and lack of experimental comparison led reviewers to question to what extent RAEE represents a significant innovation.

**Reviewer Concerns:**

The authors commented on all of the concerns raised by the reviewers. Concerns around theoretical underpinnings (Jeffreys–Beta prior), the "comparative trap" statistical vs cognitive biases, were sufficiently addressed. Reviewer sUeC specifically raised concerns around the lack of comparison and innovation beyond ArenaHard Auto and other prior works. The authors agreed with some of the concerns and argued their work still represents a significant contribution, however, it's arguable that this concern still remains.

**Reviewer Scores:**

- Q8hk: I think the reviewer would have increased their score by +1 to 5 in light of the rebuttal.
- TNg3: It appears the reviewer had a chance to respond to the rebuttal, so I believe their score of 8 is accurate.
- sUeC: The reviewer said they would increase their score. I expect they would raise it from 2 to something like a 5.

---

### Decision · Program_Chairs · 2026-01-26

Reject